# Future Wintertime Meridional Wind Trends Through the Lens of Subseasonal Teleconnections

Dor Sandler[1] and Nili Harnik[1]

[1]Department of Geophysics, Porter School of the Environment and Earth Sciences, Tel Aviv University, Tel Aviv, Israel

**Correspondence:** Dor Sandler (dor.sandler@gmail.com)

**Abstract.** Large-scale atmospheric circulation is expected to change considerably in the upcoming decades, and with it, the interaction between Rossby waves and the jet stream. A common feature of midlatitude wintertime variability is upper tropospheric quasi-stationary number-5 wave packets, which often propagate zonally along the jet. These are collectively referred to as the Circumglobal Teleconnection Pattern (CTP). Their likeness seemingly emerges as a robust signal in future meridional wind trend projections in the Northern Hemisphere, which take the form of a zonal wave encompassing the midlatitudes.

We attempt to elucidate this link across timescales (daily, monthly and climatological), focusing on wave propagation in the jet waveguide in reanalysis and a 36-member ensemble of CMIP5 models. Using EOF analysis on 300 hPa subseasonal V anomalies, we first establish the ensemble's skill in capturing the pattern. Then, by investigating EOF phase space, we characterize the CTP's behavior in present day climatology and how it is projected to change. Under RCP8.5 forcing, most models develop a gradual preference for monthly-mean waves with certain longitudinal phases. The ensemble is thus divided into subgroups based on region of increased wave activity. For each model, this region corresponds to a more pronounced local trend, which helps explain the ensemble projection spread. Additionally, in two test-case models, this coincides with an increasing number of preferably phased wave packets at the synoptic scale. Some signs suggest that differences in CTP dynamics might stem from mean flow interaction, while no evidence was found for the role of tropical diabatic forcing.

Thus, we conclude that this climate change response, seemingly a single large-scale wave, is actually comprised of several regional effects which are related to shifts in CTP phase distributions. The strong dynamical disagreement in the ensemble then manifests as significantly different circulation trends, which in turn might affect projected local temperature and precipitation patterns.

## 1 Introduction

Projections of future circulation trends, driven by anthropogenic climate change, commonly display large scale patterns (Collins et al., 2013). Studying these structures in the context of changes in the climatological mean flow is essential in order to understand the underlying basic dynamical mechanisms. However, as some signals follow oscillating intrinsic modes (Branstator

and Selten, 2009), there are certain benefits to tracing their transient development on a higher temporal resolution. Namely,
tracking changes in subseasonal-to-seasonal fluctuations that might have important societal impacts for certain regions.

One example of such a pattern is the Circumglobal Teleconnection Pattern (CTP), first defined by Branstator (Branstator 2002, hereafter B02). The term describes quasi-stationary Rossby waves in the upper-troposphere, which are zonally oriented and span the Northern and Southern Hemispheres (NH and SH respectively). On a subseasonal scale, one can picture the CTP as a "family" of related patterns, all of them waves with an arbitrary longitudinal phase. As they are quasi-stationary
and equivalent-barotropic, this means that lobes of anomalous meridional flow with near-zero phase velocity can prevail over specific regions for several days, sometimes inducing extreme weather such as cold spells (Harnik et al., 2016) and precipitation extremes (Feldstein and Dayan 2008; Teng and Branstator 2017).

Each wave zonally circumscribes the globe along a narrow latitudinal band in the area of the climatological tropospheric jet stream, which serves as its waveguide. The characteristics of these waves (such as scale and amplitude) vary between different
regions and seasons (Branstator and Teng, 2017). They are locally determined by the interplay of the jet, its perturbations and at times remote diabatic forcing, most notably in the form of tropical convection (Yasui and Watanabe 2010; Yuan et al. 2011). Through linear Rossby wave theory, one can show that the wintertime (December-February, DJF) boreal waveguide favors stationary number-5 waves, which are at the center of the CTP definition.

In this work we focus on the wintertime CTP, but a summertime variant exists as well. In the boreal summer, the NH jet
stream shifts poleward and is typically weaker. Therefore, the stationary waves associated with the CTP are shorter in scale (mostly $k = 6$) and lifetime (Branstator and Teng, 2017). Nevertheless, the summertime CTP was also found to be related to extreme weather, such as heat waves in Southeast China (Wang et al., 2013) or extreme precipitation over western Europe (Saeed et al., 2014).

One method commonly used for isolating these patterns is Empirical Orthogonal Functions (EOF) analysis. In B02, the
45 author calculated the two leading EOFs of the subseasonal anomalies of monthly 300 hPa meridional wind for the entire NH. This means that each season's mean was removed from every monthly mean $V$ map. The resulting patterns, a pair of rather similar number-5 waves in quadrature, were found to be very robust. B02 showed that they can be produced from different fields' EOFs, as well as other methods altogether (like one-point correlation). Further explanations of the EOFs used in this study are provided in Sections 2 and 3.1.

Mathematically, the subseasonal manifestation of the CTP is constructed from the linear combination of the two leading $V$ EOFs:

$$\boldsymbol{P}(\gamma) = \cos(\gamma)\boldsymbol{V}_{EOF1} + \sin(\gamma)\boldsymbol{V}_{EOF2} \tag{1}$$

Where $\gamma$ signifies the phase for the resulting wave pattern $\boldsymbol{P}$. This simple deconstruction captures the longitudinal CTP
phase, which is an important yet often overlooked characteristic. In B02, regional EOF analysis was performed in order to determine the frequency and teleconnectivity of differently-phased waves. It was concluded that the waves have no preferred

longitudinal phase, meaning that combinations of the patterns have a uniformly distributed probability distribution function (PDF). This result was later expanded upon in a 1800 year global climate model (GCM) preindustrial control run (Teng and Branstator, 2017).

Another useful perspective for CTP analysis is through Rossby wave packets (RWPs). These RWPs, sometimes called wave trains, are local synoptic structures. The ones that are associated with the CTP are typically comprised of one or two wavelengths, seen as a sequence of troughs and ridges in geopotential height, or as alternating northward-southward anomalies in meridional wind. One noteworthy feature of these specific RWPs is their velocity. They have positive zonal group velocity and near-zero phase velocity. So while their centers of action remain almost stationary, the envelope propagates eastward

along the jet. This creates the impression of new stationary troughs/ridges appearing downstream of the original RWP, dubbed "downstream development" (Orlanski and Chang, 1993).

  This makes the CTP a unique bridge between timescales. The prevalence and quasi-stationary nature of these synoptic RWPs allow their influence to manifest on a subseasonal scale as well. The interpretation of the CTP as the mean signal of many RWPs was suggested by Watanabe (2004), who examined the packets generated after North Atlantic Oscillation (NAO)

events in both interannual and intraseasonal time scales. Feldstein and Dayan (2008) later showed the high spatial correlation between RWPs traversing Europe and the Mediterranean and the leading global $V$ EOF, calling the former "a fundamental pattern of variability in the Northern Hemisphere". A similar approach was utilized in Harnik et al. (2016) to link the second global EOF to RWPs crossing the Pacific to North America.

  Multiple processes can influence the generation and development of RWPs (see section 4b in Wirth et al. 2018). Excitation

of a RWP requires, first and foremost, an initial perturbation near the waveguide. These involve either diabatic latent heating (due to mesoscale convective systems, for example) or potential vorticity anomalies (originating from the lower stratosphere or interacting separate waveguides), as was explored in Röthlisberger et al. (2016). Specifically, the NAO can act as a precursor to such RWPs, affecting their path and amplitude (Watanabe (2004); Wolf et al. (2018)).

  It has been previously posited that the CTP, as a low frequency internal mode, will be excited more frequently in the future, as

greenhouse gas (GHG) forcing grows stronger (Branstator and Selten, 2009). Multiple experiments indeed revealed a CTP-like trend in upper tropospheric flows in response to an increase in GHGs. This is true for both the CMIP3 and CMIP5 ensembles, using multiple runs of a single model (Selten et al., 2004) or a multimodel dataset (Brandefelt and Körnich 2008; Simpson et al. 2016).

  Most notably, Simpson et al. (2016) found a robust zonal number-5 wave structure in the 300 hPa eddy meridional wind

trend under the RCP8.5 scenario for 35 CMIP5 models. This is accompanied by increased presence of stationary number-4 and number-5 waves at the expense of shorter ones ($k \geq 6$), as well as robust precipitation anomalies over North America. It was determined that mean flow changes, and specifically future acceleration of the jet, were the driving force of this scale shift. Wills et al. (2019) expanded upon these results and emphasized the eastward phase shift of wintertime stationary waves caused by the future wavenumber changes. For the summertime waveguide, the driving mechanism might be more complex, as signs

show that changes in the diabatic forcing (convective heating, land interactions) play an important role in the amplification of the teleconnection (Teng and Branstator, 2019)).

Some aspects of these teleconnection trends are still not well understood, however. Many of the aforementioned studies highlight the considerable uncertainty in the regional amplitude of the ensemble's response. For example, Simpson et al. show that model differences in present wave amplitude climatology and future jetstream acceleration can together explain 50 % of the intermodel trend variance. Additionally, on the subseasonal scale, the variance of quasi-stationary waves (calculated using 300 hPa zonal $V$ anomalies) is actually projected to decrease, in an apparent contrast to the seasonal signature (Wills et al., 2019).

There is a conceptual gap that complicates the establishment a direct causal link between the CTP and the wavy number-5 trend found in climate change projections. Namely, how do changes in jet driven subseasonal variability translate to long-term climatological shifts? In this paper, we aim to elucidate this connection, which spans across time scales, between future long-term circulation trends and jet driven natural variability. Despite the "Circumglobal" part of its name, throughout this work we use the term CTP in its most general sense - a class of regional wavenumber-5 patterns, rooted in the synoptic scale while showing a clear signature in the subseasonal range and over the entire hemisphere.

We hypothesize that the robust wavenumber-5 climate signal is brought about by changes in the waves that comprise the CTP. It is then possible that the key to understanding the trend lies in the statistics and dynamics of wave propagation in the jet waveguide, most crucially at the synoptic and subseasonal scale. These shorter temporal scales might also imply that this hemisphere-wide climate change signature is actually made up of several different regional effects.

After providing data descriptions and definitions (Section 2), we address the issue of how well the CMIP5 ensemble captures the CTP (Section 3.1). Then, we characterize the CTP's behaviour in present day climatology and how it changes as GHG forcing intensifies throughout the $21^{st}$ century in Section 3.2. After differentiating distinct CTP responses in the ensemble, we compare model subgroups and daily resolution test cases (Section 3.3) in order to better understand the causes and nature of these changes to the pattern. This is followed by discussion and conclusions in Section 4.

## 2  Data and methods

### 2.1  Observational data

In order to estimate how well the CMIP5 ensemble captures the CTP, we first turned to reanalysis. The analysis is performed using monthly and daily mean data from the National Centers for Environmental Prediction I (NCEP-I; Kalnay et al. 1996) and ERA-Interim reanalysis (Dee et al., 2011) datasets. We focused on the winter season of December to February (DJF), between 1958-2015 for NCEP-I and 1979-2014 for ERA-Interim. For direct comparison, the ERA-Interim data (as well as all CMIP5 model output described later) was interpolated to fit the NCEP-I $2.5° \cdot 2.5°$ horizontal grid. Unless stated otherwise, the circulation is examined in the 300 hPa meridional wind field ($V$) between $10° - 85°N$.

Following B02, we generated meridional wind EOFs using monthly subseasonal anomalies of $V$, meaning that each winter's DJF mean was removed from its monthly mean $V$ fields. Subseasonal anomalies are used in order to negate large scale interannual signals (driven by SSTs for example) which otherwise dominate the EOFs. Meridional wind is chosen instead of streamfunction or geopotential in order to better capture zonally elongated, smaller scale patterns such as the CTP (see Ap-

pendix A in Branstator and Teng 2017). While B02 uses non-divergent $V$, we employ the full field. As noted in Harnik et al. (2016), this does not affect the resulting patterns.

As for the spatial domain, "global" EOFs are calculated using all longitudes; and regional EOFs are based on $144°$ longitudinal sectors (roughly equals two number-5 wavelengths). The Euroasian domain (denoted as AS) spans $0° - 144°E$, and the Pacific North American (NA) domain covers $180° - 324°E$. These particular regions were chosen due to their position right

downstream of areas in the North Pacific and Atlantic oceans with considerable Rossby Wave Packet activity during winter months (Souders et al. 2014; Röthlisberger et al. 2018). The EOF analysis was employed on these domains separately (as opposed to simply using sectors of the global EOFs) in order to preserve the orthogonality of the functions. Harnik et al. (2016) showed that the regional patterns are well matched with the corresponding sector of the global ones.

## 2.2  CMIP5 runs

For model data, we use simulations from 36 GCMs in the CMIP5 ensemble (Taylor et al. 2012; for the full list of models used, see supplemental data). Monthly mean flow fields (meridional and zonal wind) are analyzed for all models on the 300 hPa level, similar to the reanalysis data. We also investigate these fields (as well as 250 hPa horizontal divergence and outgoing longwave radiation) in the daily resolution for two models (IPSL-CM5A-MR, MIROC-ESM-CHEM) as representative test cases for the two dominant types of climate responses that were observed in the ensemble. The daily mean flow, however, is

taken from the 500 hPa level, as 300 hPa data was not publicly available for these daily runs. A comparison between the two levels (not shown) reveals that the two are adequately similar in terms of the EOFs and associated CTP circulation. This is to be expected, as the CTP is considered an equivalent-barotropic phenomenon.

All models are examined under the Historical and RCP8.5 scenarios. Historical runs span the period between 1900-2005 in the monthly resolution and 1950-2005 for daily means. RCP8.5 runs are between 2006-2099 for all temporal resolutions.

For one daily data test case (IPSL-CM5A-MR), a longer Pre-industrial Control run was also used (1800-2100). The EOFs for every model were produced in a similar manner to the reanalysis datasets (for the hemispherical and two regional domains), using Historical data only. For the most part, longer datasets are preferable for this type of calculation. However, we opted to exclude Future data (2006 and onward). This allows better comparison between models and reanalysis, while also keeping the distinction between the studied pattern and the projected climate signal which might affect its behaviour.

## 2.3  Pattern correlation and EOF phase space

Pattern correlation is used extensively throughout this work as a tool for comparison between models and observations, and for projection scores on the EOF phase space (as explained below). We calculate the Pearson correlation coefficient for two maps, each weighted by the square root of cosine latitude. It is important to note that this measure is useful in assessing pattern similarity and phase difference, but does not reflect amplitude differences (for matrices $\boldsymbol{A}, \boldsymbol{B}$ and scalar $c$, $PCC(\boldsymbol{A}, \boldsymbol{B}) =$

$PCC(\boldsymbol{A}, c\boldsymbol{B})$ ).

We define the CTP as any linear combination of the leading two EOFs, encompassing all phases. Therefore, in order to quantify how "CTP-like" a certain flow pattern $P$ is, we construct a phase space based on EOF1 and EOF2. The complex

projection score of map $\boldsymbol{P}$ is thus $PCC(\boldsymbol{P}, EOF1) + iPCC(\boldsymbol{P}, EOF2)$, revealing both the level of similarity between the flow and the CTP, and the longitudinal phasing of the pattern. The former corresponds to the modulus of the projection score and the latter to its argument (Fig. 1).

The question of what kind of data to project on the EOF phase space is not a trivial one. For the daily time scale, the progression of Rossby wave packets can be properly captured by the full $V$ field, achieving projection scores of 0.5 and higher. Additionally, the CTP is essentially a subseasonal phenomenon and is thus expected to have a signature in data with longer timescales as well. Nonetheless, full field monthly data projections generally fall short in terms of pinpointing the pattern. This might be due to planetary-scale stationary waves which are more prominently featured on those timescales. We thus use $V$ anomalies, or deviations from a 30 year climatology. Considering that the climatology is itself changing throughout our experimental period (from pre-industrial times to the late $21^{st}$ century), we chose early $20^{th}$ century (1900-1930 DJF mean) as a representative reference period, which we then remove from the monthly mean $V$ field. Our results were found to be insensitive to the selection of climatological time span.

After excluding data with low projections scores (below the $70^{th}$ percentile of index magnitude), we average all remaining monthly values to obtain the mean projection score of every experimental run. Patterns with an overall arbitrary phasing will have oppositely signed monthly scores cancelling each other, and therefore a mean projection close to the origin. When a model's mean score exceeds the RCP8.5 Multi-Model Mean (MMM) projection score by one standard deviation, it is considered to have a "preferred" phase. A model subgroup, used for trend and forcing comparisons, is defined by all models whose preferred phase over a geographical domain (NA or AS) occupy the same half-plane on the phase space.

Statistical significance of group composites is determined by a sign test, which indicates where a certain percentage of group members have the same sign as the composite mean. The chances for a given percentage of events to have the same sign as the composite mean are determined using a binomial formula, assuming equal chances for positive and negative anomalies.

### 2.4 CTP events

We define "CTP events" in daily mean data. These are essentially Rossby Wave Packets that are nearly in phase with the 500 hPa equivalent of the preferably-phased pattern that was found in the 300 hPa monthly projections. First, we apply a 3-day running mean on the 500 hPa daily $V$ field. After calculating the projection index and excluding low values (as in the monthly case), we detect all sequences of three or more consecutive days in which $|\phi_d - \phi_m| \le \pi/8$, where $\phi_d$ and $\phi_m$ are the daily and preferred-monthly phases respectively.

This approach is focused on the phase of the RWP, but the use of EOFs also captures the meandering character of their path. Alternatively, one can possibly further filter the wind field and use only the component perpendicular to the background flow (Wolf and Wirth, 2017).

When creating wave composites, we remove the Future climatology from the daily means in order to observe wave propagation and to remove the signature of other low frequency patterns. Lag 0 of a CTP event is defined as the first day of the sequence and statistical significance is determined by a 1000 member bootstrap method.

To test the NAO's possible role in forcing the CTP, we use a standard NAO index (Hurrell et al., 2003): projection of monthly sea-level pressure (SLP) anomalies onto the leading EOF of the seasonal SLP anomaly in the North Atlantic sector ($20°N − 70°N; 90°W − 40°E$). Positive and negative NAO phases are defined as months when the index exceeds one standard deviation ($±σ$).

Additionally, we used lagged linear regression (Livezey and Chen, 1983) of outgoing longwave radiation (OLR) and upper tropospheric divergence, in an attempt to establish a causal relationship between tropical convective forcing and the excitation of CTP events. However, the resulting patterns were not statistically significant, hence additional technical details are only provided in the supplementary materials.

## 3 Results

### 3.1 Representation of the CTP in the CMIP5 ensemble

The CTP can be obtained by calculating the first two leading EOFs of the monthly winter (DJF) subseasonal anomalies of the upper-tropospheric meridional wind. The observational patterns for NCEP-I reanalysis can be seen in Fig. 2. Very similar patterns were also produced from ERA-interim data (not shown).

In both cases, the set of EOFs is comprised of two quasi-stationary zonal number 5 waves which are in quadrature with one another. They explain 13.6 and 10.8 % of the variance (denoted by $λ$) in the NCEP/NCAR dataset, and 13.3 and 11.5 % in ERA-Interim. For ERA-Interim, the two EOFs are not well-separated from one another (as well as from the third EOF) according to the definition set by North et al. (1982). For NCEP-I, a longer dataset, the three leading patterns are well-separated from each other.

Performing the same calculations on 36 GCMs from the CMIP5 ensemble reveals the robustness of this pattern. In order to allow a comparison to observational present-day climate, the monthly data used in calculating the EOFs came from historical model runs only (between the years 1900-2005). We base all of our calculations and projections hereafter on the historical EOFs, even when working with data from Future runs, as no considerable differences were found in EOFs based on RCP8.5 data (not shown).

Figure 3 demonstrates the strong agreement between models in regards to the main features of the CTP (for every individual model's EOFs, see supplementary Fig. S1). Most notably, almost every model has zonal number 5 waves in their two leading EOFs, with a stronger amplitude above North America and correctly phased patterns. As expected from theory, the waves are latitudinally confined to the area of the climatological jet stream. However, it is worth noting that model agreement on the first EOF is more robust, and that it is also closer to the reanalysis-based function, as will be demonstrated quantitatively.

For most models, the variance explained by these patterns is slightly higher than in reanalysis, with median ensemble values of 14.8 % for the first EOF, and 10.7 % for the second. Of the 36 ensemble members, 28 have their two leading EOFs well-separated from the third function (see Table 1). This might be a result of the model runs being almost twice as long (105 winters) as the observational records (57 winters in the longer dataset, NCEP-I).

In order to quantify each model's skill in representing the CTP, each set of two EOFs was projected onto the observational patterns with cosine latitude weighting. Most models can reproduce the spatial features of the CTP fairly well. Zonal number 5 patterns will have a large absolute score (1 for a perfect copy, -1 for the same wave in antiphase), and a zero score represents some flow unrelated to the CTP. The ensemble has a median score of 0.78 and 0.5 for the first and second EOFs respectively. There is however a larger spread in skill for the second function.

## 3.2 Changes in CTP behavior in a warming world

### 3.2.1 Projecting the MMM climate trend onto the EOFs

A quasi-stationary zonal number 5 wave in the northern hemisphere appears to be a prominent feature of the projected circulation trend in high emission scenarios. The similarity between the CTP and CMIP5 future circulation trends has been noted before (Brandefelt and Körnich 2008; Branstator and Selten 2009; Simpson et al. 2016). However, to our knowledge no effort has yet been made to objectively link the two based on the EOF perspective that was used by B02 to first define the CTP.

Under scenario RCP8.5, the multi-model mean response in the 300 hPa meridional wind indeed takes the form of a number 5 wave globally (Fig. 4). This response being the Future climatology (2070-2099) minus the Past climatology (1979-2005), similar to the one analyzed in Simpson et al. (2016). Despite the apparent similarity, the pattern itself is not a pure combination of the global EOFs. When projecting each model's trend onto its historical EOFs, one gets rather low scores (absolute mean of 0.24 and 0.21 for EOF1 and EOF2).

This issue can be solved by dividing the trend into different regions. By doing so, we can inspect whether separate longitudinal sectors in the northern hemisphere develop different CTP-like responses. The regions used, as elaborated in Section 2.1, are Pacific-North America (NA) and Euroasia (AS), each spanning $144°$ longitudinally, or approximately 2 wavelengths (see boxes in Fig. 4a). Calculating the EOFs anew, as opposed to using segments from the global ones, makes sure that the functions remain orthogonal (the MMM regional EOFs are presented in supplemental Fig. S2). As was the case for the global EOFs, the regional EOFs display a good resemblance to their observational counterparts, both in qualitative structure and pattern correlation score (not shown).

In order to confirm that the regional EOFs are more suited for describing the trend, we calculated how much of its spatial variance was explained by the EOFs (Fig. 4). This was done individually for every model, as well as for the multi-model mean (using composite EOFs and the MMM trend). For most models used, the variance is spread quite evenly across the first five EOFs (the higher-ordered of which have lower $\lambda$ values). In particular, the first two leading functions, which define the CTP, explain less than half of the trend's variance for 26 of the 36 ensemble members. As for the two leading composite EOFs, they account for only 0.8 % of the MMM trend variance, while the majority is explained by the third and fourth composite functions.

When looking at the spatial variance explained by the regional EOFs (Figure 4c-d) one can see that, for the majority of the ensemble, they are better at describing each model's trend. The ensemble median value for the variance explained by the first two leading functions is 61 % for the NA region and 56 % for AS. This is almost twice as the median for the global EOFs,

which explain 32 % of the trend. This allows us to use only these regional functions for our analysis (two EOFs for each region).

This improvement might seem unexpected, considering the spatial similarity between the global and regional EOFs (see Fig. S2 in the supplementary material). However, this is by virtue of the added degrees of freedom (an index consisting of four regional projections, instead of two global ones).

### 3.2.2 Emergence of preferred phasing in monthly data

With some association between the trend and the CTP established, it is important to understand the temporal development of this signature. The data being used in this section is the monthly DJF 300 hPa meridional wind, for both Historical and RCP8.5 runs in each model. The Historical run spans from 1900 to 2005, and RCP8.5 from 2006 to 2099 (adding up to 315 and 279 months, respectively). From these maps we then remove the pre-industrial winter climatology (1900-1930).

Using the EOF1-EOF2 phase space thus reduces the relation between each month and the CTP into common two dimensional vector parameters - magnitude (Euclidean norm of both projection scores) and angle. The former reveals how much does each month resemble the CTP and the latter signifies what longitudinal phase it has.

For the projections, we use the two sets of regional EOFs. After removing months with low projection scores, we examine the remaining distribution across the phase space, and also compare it to NCEP-I data. For all 36 models, historical data is spread approximately uniformly on the phase space (blue dots in Fig. 5, shown for two single model examples), meaning that CTP phasing is arbitrary throughout these runs. This is similar to the spread in reanalysis data (not shown) and previous studies (Teng and Branstator, 2017).

However, this is not the case for the projected future. Under the RCP8.5 scenario, most models show a preference for certain phases, for either the NA or AS regions (or both, in the case of two specific models). This means that the number of winter months that fall within a specific quadrant on the phase space is significantly higher than what would be expected from a uniformly distributed dataset. This trend becomes even more pronounced towards the end of the $21^{st}$ century. Projections on global EOFs do not reflect these results, but rather remain uniformly distributed throughout all runs (not shown). This is due to the regional scale of the emerging patterns. For example, months with a good resemblance to the North American sector of the CTP might still receive a low projection score due to a lack of wave activity over Euroasia.

The mean location of every model on the phase space illustrates this idea of preferred phasing (Fig. 6). A uniform distribution of monthly projections, centered around the origin, will trivially yield a mean projection score close to zero (as is the case for all Historical CMIP5 runs). In contrast, more than half of the ensemble's RCP8.5 runs have a mean projection score bigger than $1\sigma$ for at least one region ($\sigma$ being the standard deviation of all RCP8.5 mean projection scores). For the NA region, nine models have a majority of months with a positive (negative) time-mean score for EOF1 (EOF2). Eight models display a preference in the AS region, with negative (positive) scores for the regional EOF1 (EOF2). Two additional models show these tendencies for both domains.

Interestingly, we can identify a common phasing that is shared by all strongly projecting models in each region (in Fig. 6, these are the filled red rectangles which are outside the circle). Specifically, almost all models that have a non-zero mean pro-

jection onto the NA phase space concentrate around approximately $\gamma = -\pi/8$. Similarly, seven models with mean projections on the AS EOFs are in and around the second quadrant. This allows us to subjectively define four distinct model subgroups (NA, AS, BOTH, NONE) according to their region of preferred phasing (Table 1). The preferred patterns themselves can be obtained by following Eq. (1).

Members of the NA group have an abundance of months with a ridge located near and off the west coast of North America. Teng and Branstator (2017) noted that there is "a continuum of low-frequency zonal wavenumber-5 patterns" which can produce this structure. The preferred phasing shared by the NA models (a positive EOF1 and a negative EOF2) is indeed included within this continuum. Meanwhile, the AS group's models show a typical monthly anomaly consisting of a strong northerly flow across the Eastern Mediterranean and India, and southerly winds over the Arab peninsula and Southeast Asia. This is quite similar to the positive phase of the Southern Levant (SL) pattern, as defined by Feldstein and Dayan (2008). The combination of these regional flow patterns is very reminiscent of the MMM climate trend, as seen in Fig. 7.

We attempt to clarify the connection between the subseasonal pattern and the climatology by examining the different model subgroups. There is strong agreement within the ensemble in regards to the general structure of the 300 hPa meridional wind trend between the Future and Past periods. However, as previously mentioned, there is considerable spread in the magnitude of the response, with up to a 5 ms$^{-1}$ difference in wave amplitude for some regions.

By dividing the models based on their region of increased CTP activity, significant local differences in the climate trend become apparent (Fig. 8). For every group (other than the NONE group of course), the region with preferably phased waves also shows a stronger trend response. Three areas stand out in particular as sources of disagreement between groups: The AS group has a more pronounced signal over the Mediterranean and South Asia (boxes 1-4 in Fig. 8c) and around the Bering Sea in the northern Pacific (boxes 5-6). As for the NA group, its stronger signal is found over North America (boxes 7-9), as expected. Fig. 8d illustrates this point, as regional wave amplitudes of the trend differ between model groups. This result expands on Brandefelt and Körnich (2008), who also identified analogous local trend differences in a smaller set of the CMIP5 ensemble.

## 3.3 The wave packet perspective - CTP in daily data

### 3.3.1 The daily data phase space

Using monthly data is limiting in terms of analyzing the actual progression of the waves, their duration or temporal frequency. In this section, the dynamics of the waves will be explored using daily mean $V$ data from two members of the CMIP5 ensemble - IPSL-CM5A-MR and MIROC-ESM-CHEM (abbreviated below as IPSL and MIROC). The latter is included in the AS group and the former in the NA group, and both are in the ensemble's top $20^{th}$ percentile in terms of spatial correlation to observational EOFs.

Using 500 hPa EOFs and daily mean $V$, we can again calculate projections of the data and get the score and phasing for the Historical (1950-2005) and RCP8.5 (2006-2099) runs. The daily phase preferences are close to the monthly ones in terms of angle ($|\phi_{day} - \phi_{mon}| < \pi/4$). However, this does not mean that the two timescales capture the same phenomena, as correlation between the two is non-trivial. This will be further inspected in the Section 4.

### 3.3.2 CTP events

Next, we define the "CTP event". These events provide a way of capturing the teleconnection in the form of RWPs, as it is manifested in synoptic timescales. Essentially, a CTP event is a sequence of at least three consecutive daily timesteps with high projection scores and angles close to the preferred mean monthly phasing. These events are defined in relation to a specific region, according to the EOFs used for projection. The list of potential events is further filtered in order to remove false positive matches (for additional details, see Section S1 in the supplementary material).

We first apply this method to reanalysis, with wave phases taken from the CMIP5 subgroups for the NA and AS regions. For the NCEP reanalysis, an average of 1.5 events per season was found in the NA region, and 0.5 events per season in AS. Meanwhile, the ERA-Interim dataset seems to capture more events, with an average of 1.8 and 0.9 per winter for the two respective domains.

For comparison, Souders et al. (2014) found that, on average, about 11 RWPs form over the Pacific Ocean every winter, while 9 are formed over the Atlantic. This was calculated by tracking $\sim 6000$ RWPs that appear globally throughout the year. Note that the comparison to our results is not straightforward. For instance, a RWP formed in the Pacific might be categorized in both of our groups (depending on its path). Additionally, our CTP events likely constitute only a small subset of this climatological RWP dataset, which covers all phases and a wide range of wavelengths. This makes their result serve as a loose upper bound. Another more specific bound can be found in a work by Wolf et al. (2018). The authors examined the climatology of long-lived quasi-stationary waves using the ERA-Interim dataset, and found an average of 0.6 instances per winter (DJF) of waves with a lifetime of 10 or more days. This is more directly related to circumglobal RWPs, which typically require more than a week to complete their path.

In the two GCMs, behavior of the CTP changes between the Historical and RCP8.5 periods, much along the lines of the monthly data results. For IPSL and the NA region, the number of events nearly doubles between the Past and Future periods (from 1.6 to 3 events per winter). In the AS region, MIROC CTP event frequency increases from 1.1 to 1.9 for the same metric. It's important to note that these changes only happen for a narrow range of wave phases (less than a full quadrant) within every model's domain. One interesting exception is found in MIROC, which shows this trend for both domains, in the daily timescale only. The average lifetime of the wavepackets does not change between the runs, and was found to be 6 and 5.5 days for IPSL and MIROC, respectively.

For both models, the additional events are not evenly spread throughout the RCP8.5 winters, but are rather concentrated in a few seasons with increased wave frequency (Fig. 9). As the average number of events per winter increases, the tail of the distribution shifts as well. Thus, we begin to see winters with five or more events, which is unprecedented before 2006. Performing the same analysis on a long Pre-industrial Control run (unfortunately available only for IPSL) further underlines how the projected RCP8.5 future differs from unforced natural variability.

### 3.3.3 RWP propagation

We can mark the first day of each series of CTP days within an event as lag 0 and follow the propagation of the RWPs. By definition, each such lag 0 day has a high projection score on the preferred phase, meaning that the RWP is already past its initial excitation and is at least somewhat developed. By creating a simple composite of all events for each model and domain, we observe the general features of a "typical" RWP, without tracking individual waves.

These composite wave packets (seen in Fig. 10) display realistic characteristics that have been previously affirmed in re-
360 analysis and models (Souders et al. 2014; Röthlisberger et al. 2018; Wirth et al. 2018): They are of synoptic scale and their propagation takes place over the course of approximately one week. Their centers of action remain fairly stationary through-out, meaning that they indeed have near-zero phase speed. This is by construction, as the RWPs were chosen to have persistent strong projections onto the same phase. The waves propagate zonally by downstream development, with a new positive or negative anomaly forming every 48 hours or so. Furthermore, the waves in both models are excited over regions that have been
previously identified as prominent sources for winter RWP activity.

For the IPSL case, the waves' source is located over Southeast Asia. They traverse the Pacific Ocean and on its Eastern coast they project strongly on the positive phase of the Pacific-North American teleconnection (PNA). Similar to the monthly case, this longitudinal section has the strongest response, with an amplitude of up to 5 m s$^{-1}$.

On the other hand, the Euroasian waves in the MIROC run originate near the United States. The regional EOF projection
captures a mature stage in their propagation, as lag 0 (Fig. 10f) occurs about a week after the initial excitation. Interestingly, this marks only an intermediate point along their near-circumglobal trajectory. After crossing the Atlantic, the composite RWP propagates further across the Pacific and finally projects onto the negative phase of the PNA, near the wave's origin.

Over this region, the later part of the AS-projected wave is in antiphase with the NA-projected one (compare North America in Fig. 10g,j). This might explain why MIROC shows no monthly preferred phasing in the NA region. Despite having the same
trend in daily data as IPSL (i.e. more frequent RWPs with $\gamma \approx \pi/8$), this activity is masked by waves of opposite phase when averaging on longer timescales.

Since the composites are based on EOFs (which are also not well-separated), we need to carefully confirm that this circum-global signature is not just a mixture of different local wave patterns in various stages of propagation. Indeed, over 60 % of individual CTP events have a strong projection ($\geq 0.2$) onto both early and late stages of the composite propagation (lag -4
and +4, respectively), with significant wave activity across both oceans. Furthermore, Southeast Asia has been found to have far-reaching teleconnectivity, with high point-correlation to centers around the globe (B02). This alludes to the fact that these RWPs might be truly circumglobal.

In this light, the spatial structure of the MMM trend can be understood in greater detail. It seems that most anomalous centers can be related to one of the wavetrains that become prevalent in the projected future in daily data. Specifically, boxes
1-4 (7-9) in fig. 8c correspond to the wave in fig. 10f (10g). Over the Pacific region, it is possible that the North-South dipoles are actually a combination of the two RWP behaviors found in the ensemble (Fig. 10g,j).

## 3.4 Differences in the mean flow and forcing

While differences in CTP activity might shed some light on the spread in the climate response, the question still remains: What causes these differences in the first place? One element which might allude to the answer is the mean flow. The zonal mean flow, a crucial factor in the propagation of RWPs, displays some variance within the ensemble. It appears that for each group, in both Historical and RCP8.5 runs, the CTP-prone region also has a stronger, narrower mean jet compared to the remaining ensemble mean (Fig. 11). This difference is mostly noticeable over maritime areas, just downstream of the RWP origins. It takes the form of a meridional tripolar pattern around the jet core. For the NA region, this tripole is also evident in the composite daily jet anomaly during CTP events (not shown).

As for other large scale modes which can trigger the CTP, the phase of the NAO was found to be correlated with the occurrence of monthly CTP preferred patterns in the AS region (a pattern similar to the "downstream extension" found in Watanabe (2004)). For seven models of the AS group, we calculated the share of preferably phased Future CTP months to happen concurrently with a positive or negative NAO phase. For six of the seven models, the monthly AS CTP pattern is at least twice as likely to occur with a positive NAO phase (Fig. 12). No such relation was found for preferred NA phases (not shown).

Another useful area of inquiry is the location and magnitude of tropical and subtropical diabatic heating. This forcing can, in some cases, provide the initial perturbation for the excitation of the waves. Hence, recurring heating in a specific area might help explain the abundance of RWPs with certain phases.

However, lagged regression analysis did not reveal a straightforward connection between CTP events and supposed tropical precursors. We calculated these relations for various fields and indices that are indicative of CTP activity, focusing mostly on meridional velocity or the CTP index (phase angle of strongly projected days, index magnitude of preferably-phased days). While some convective patterns emerged (combined negative OLR and positive upper-level divergence; not shown), none were statistically significant. In fact, most tropical gridpoints showed little to no correlation (<0.1) to fields related to CTP Events. This point will be discussed later on.

## 4 Discussion and Conclusions

In this study we examined how changes in the subseasonal teleconnection variability are linked to long term circulation trends in boreal winter. This was done by identifying and characterizing the CTP in 36 members of the CMIP5 ensemble, comparing present and future climates (the latter represented by the high emission RCP8.5 scenario). The ensemble was found to have decent skill in capturing this pattern compared to observations, in terms of both spatial structure and percentage of DJF variance explained (Fig. 3).

The primary finding of this work is that the majority of GCMs project the CTP to develop a preferred longitudinal phasing over time (Fig. 6). While this change is local in nature, its effect is seen on larger scales (both spatially and temporally). Specifically, this phase preference strongly corresponds to each model's future trend in meridional wind. This is in contrast to what is known from observations, where CTP phasing seems to change arbitrarily on an intraseasonal basis. This means

that for the projected future, a growing number of winter monthly mean flows take the form of a number-5 zonal wave with a specific phase. Additionally, in two studied test cases, this translates to an increasing number of preferably phased RWPs per season (Fig. 9).

This phase locking occurs on a regional scale, and there is strong disagreement in the ensemble regarding where exactly it will take place. Nine models (NA group) predict an emergence of preferred CTP phasing over the North Pacific and North America, while another 8 have a strong signal over the Euroasian region (AS group). Two more models display some mixture of these responses, and the remaining 17 show no preferred phasing at all. The future trend of meridional mean flow is projected to change accordingly (Fig. 8). NA models show a localized large-amplitude signal over their region of preferred phasing, while AS models have a nearly circumglobal wave trend with particularly stronger anomalies over Europe and the Mediterranean.

Our results are mostly in good agreement with previous works, which focused on either the pattern itself or on the projected circulation changes. Relevant climatological features of the jet stream waveguide are present in our findings, such as areas of prominent RWP initiation (North Atlantic, and the Western and Central Pacific; Röthlisberger et al. 2018) and specific phases with extended teleconnectivity (Branstator (2002)). In terms of the pattern itself, the two regional responses that we found resemble observed local manifestations of the teleconnection, named the South Levant pattern (Feldstein and Dayan (2008)) and the Circumglobal North American pattern (Harnik et al. (2016)). Both are associated with regional precipitation anomalies over their respective domains. As for the SL pattern specifically, its positive phase (which mirrors the CTP trend) is actually linked to increased precipitation over the Eastern Mediterranean. This seemingly contradicts the robust projected drying of the Mediterranean region in the ensemble. However, it is left for future works to examine the possible relations between the CTP and these regional precipitation trends in CMIP models. Nevertheless, we hypothesize that these results would have implications on surface weather, due to the presence of these recurring, persistent types of flow.

As for the relevant climate change literature, we believe our phase-centered approach extends and contextualizes previous results. The MMM $V$ trend seen in Simpson et al. (2016) is reproduced in our analysis. Their work analyzes the conditions which drive the differences between present and future climatology, while our results "fill the gaps" and explore how these processes manifest on a subseasonal scale. A major finding of their work relates the MMM trend to a shift to longer zonal scales for stationary waves. While the strongest response happens in number-4 waves, they also present a robust signal (over 75% of ensemble members) for number-5 waves. This might be related to a stronger mean CTP signature brought about by the narrower distribution of phases.

This brings up an interesting question on the relation of different scales in stationary waves. Heuristically speaking, it's useful to think about this problem as mere downscaling - the MMM trend as an averaging of many CTP-like months, which are comprised themselves of synoptic-scale Rossby wave packets.

However, the relationship between seasonal, subseasonal and synoptic timescales is not straightforward. This is effectively demonstrated in Wills et al. (2019), who actually found a projected decrease in subseasonal (or rather monthly) meridional wind variance in the NH midlatitudes. It is pointed out that this seemingly contradicts the expected enhanced signature of quasi-stationary waves. This apparent discrepancy was also revealed in our results, as we found that seasons with a high number of wave packets don't necessarily receive a high projection score in monthly data, and vice-versa. This results in only

a moderate positive correlation ($\sim 0.4$) between the seasonal monthly projection score and the number of wave packets every season. Wills et al. cite the accelerated advection of Rossby waves as a possible reason for the inconsistencies, as their signature is smoothed out on longer timescales.

The CTP is most commonly observed and analyzed from a hemispheric point of view, and yet GCMs predict significant regional differences in wave behavior under GHG forcing. The different local trends identified in Brandefelt and Körnich (2008) partially match our categorization of NA and AS trends in the ensemble. They too noted that differences between GCMs (CMIP3 members in this case) span either the North Pacific and North America, or the Atlantic and Euroasia. A comparison can be made, to a certain degree, between the model groups and trends shown in this work and by Brandefelt and Körnich. Specifically, in both works, models from the GISS and MIROC centers were found to have a similar "Euroasian" response - poleward anomalies over western Europe and the Arab Peninsula, and equatorward ones over eastern Europe and India. There is little in common for other ensemble subgroups. This is unsurprising for several reasons. Other than the obvious difficulty of comparing ensembles from different generations, the approach used to define subgroups is quite different - "bottom-up" (subseasonal EOF projections) versus "top-down" (trend similarities).

One major issue that still remains mainly unresolved is what causes CTP phase distributions to change. We believe that understanding what drives the two very different projected CTP signals (NA and AS) might shed some light on this topic. We found significant differences in the mean flow between model groups. When compared to the ensemble mean, each group with a phase preference has an anomalous zonal flow over the relevant domain, even in the Historical climatology. Specifically, a stronger and narrower jet was identified over a sector spanning roughly one wavelength and located just downstream of RWP initiation area. Often it is difficult to differentiate mean flow properties from the resulting transient patterns. However, since this jet difference was found in the Historical model climatologies (and not in the trend itself), it alludes to a possible causal mechanism which affects the CTP.

It has been well established that such mean flow features play an important role in determining the scale and amplitude of stationary waves (Simpson et al. 2016; Branstator and Teng 2017). As far as phasing is concerned, certain longitudinal phases have longer reaching teleconnectivity (Branstator (2002)). One interpretation of our results includes a combination of both aspects. It is possible that certain local configurations of the jet stream trap longer lived RWPs from a narrow range of phases. While the overall projected strengthening of the jet leads to a robust CTP-like response across the ensemble, it is regional differences in the zonal flow which account for the model spread.

Large scale internal modes also act as forcing for the CTP. The monthly AS pattern, specifically, was found to be correlated with phases of the NAO, being twice as likely to occur during the positive NAO phase compared to the negative one. The majority of patterns, however, occur on non-NAO months (either positive or negative). This is understandable, as the NAO is only one of several mechanisms which can trigger the CTP.

This connection is in agreement with previous studies dealing with quasi-stationary wavepackets and the NAO. Wolf et al. (2018) found a strong correlation between negative seasonal NAO values and decreased wave activity in the midlatitudes. Additionally, RWPs reminiscent of the preferred AS pattern can be found in Watanabe (2004), where they are identified as an Asian extension lagging positive NAO events. It is therefore possible that, within the AS group, some common trend for NAO

variability (like a general shift towards its positive phase) forces this regional CTP pattern. This hypothesis is beyond the scope of this study.

Another possible explanation, which is yet to be supported by data, lies in the amplification of asymmetric diabatic forcing in the tropics. Areas of more frequent convection might excite more RWPs near a certain longitude, narrowing the overall phase distribution. Lagged regression analysis did not bolster this hypothesis, but rather showed a lack of correlation.

This can be interpreted in two ways: On the one hand, a lack of evidence does not irrefutably disprove causality. The CTP is only one mode of many which influence wintertime variability in the midlatitudes, and it's difficult to isolate its underlying mechanisms in the context of fully coupled models. Combined with the noisiness of convective proxies such as OLR, it might be that the correlation went unnoticed. On the other hand, the CTP is deeply intrinsic to midlatitude dynamics, and can indeed occur in the absence of forcing (Teng and Branstator, 2017). This means that diabatic heating might play a minor role in
creating the differences in wave phasing in the models.

These hypotheses require a deeper and more theoretical analysis in the future, focusing on important unanswered questions: Are local jet stream configurations driving CTP phase preferences, and what is the mechanism behind this link? Is there a trend in NAO variability that can explain the Euroasian CTP trend? What part does tropical forcing play in this, if any? And finally, how does the ensemble spread in circulation manifest in projected surface weather and extreme events?

In order to better understand the trilateral relationship between forcing, jet regimes and wave behavior, further investigation should be performed in an idealized experimental setting. This will allow a more careful analysis of fields such as Rossby Wave Source and upper tropospheric PV. The stationary wave model used in Simpson et al. (2016) is an example of such an approach. It should be interesting to see if the model can recreate different meridional flows when prescribed different asymmetric zonal basic states based on our NA and AS group composites. Additionally, more sophisticated filtering can be
applied (by wavenumber and/or through objective tracking algorithms), at the expense of generality and simplicity.

*Data availability.* CMIP5 Historical, RCP8.5 and Pre-industrial Control simulations used in this study are available from the Earth System Grid Federation (ESGF).

*Author contributions.* DS led the data analysis and writing of this manuscript. NH supervised the project, and aided with designing the calculations, interpreting the results, and writing the manuscript.

*Competing interests.* The authors declare that they have no competing interests.

*Acknowledgements.* This work was funded by the Israel Science Foundation grant 1685/17. We thank the European Centre for Medium-Range Weather Forecasts (ECMWF) for providing access to ERA-Interim reanalysis data, and NOAA Earth System Research Laboratory (ESRL) for the NCEP/NCAR reanalysis data. Additionally, we thank Ori Adam for assisting with the numerical handling of CMIP data in early stages of this project.

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

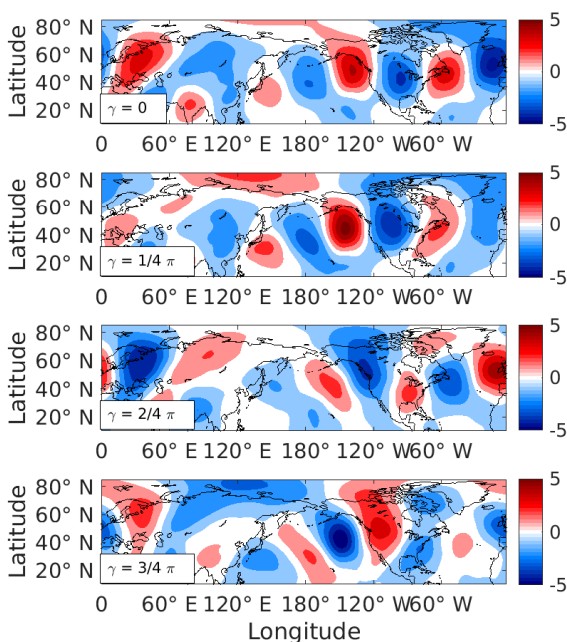

**Figure 1.** A simplified example of the relation between EOF phase angle and longitudinal phase. Each plot consists of a combination of the first two leading observational EOFs (represented by the $\gamma$ value in Eq. 1). Red (blue) shading denotes positive (negative) $V$ values.

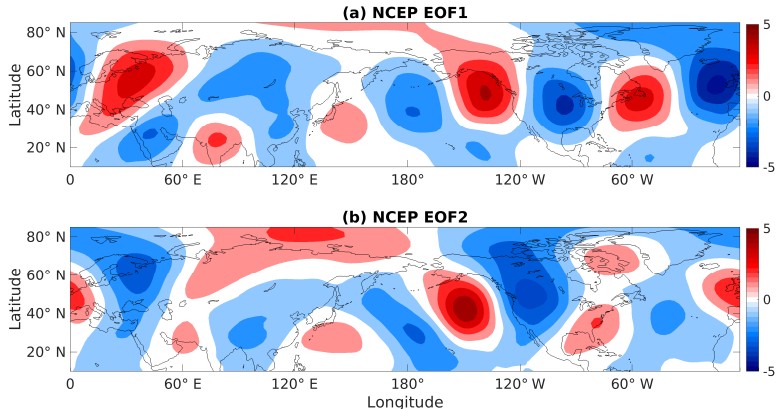

**Figure 2.** The first two empirical orthogonal functions (EOFs) of NCEP/NCAR monthly mean seasonal anomalies of 300 hPa DJF meridional wind. Red shading is positive and blue is negative here and throughout, with intervals of $1\ ms^{-1}$.

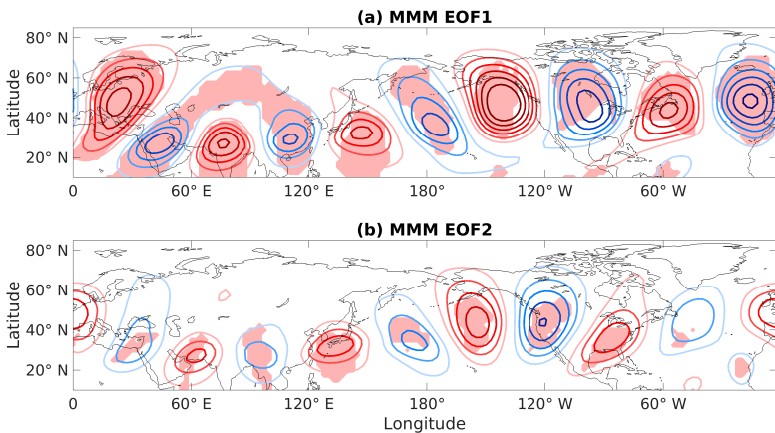

**Figure 3.** Multi-model mean EOFs for monthly mean seasonal anomalies of 300 hPa DJF meridional wind, based on Historical runs of 36 GCMs. Contour interval is 0.5 $ms^{-1}$ and shading represents areas where >90 % of models agree with the sign of the observational EOFs.

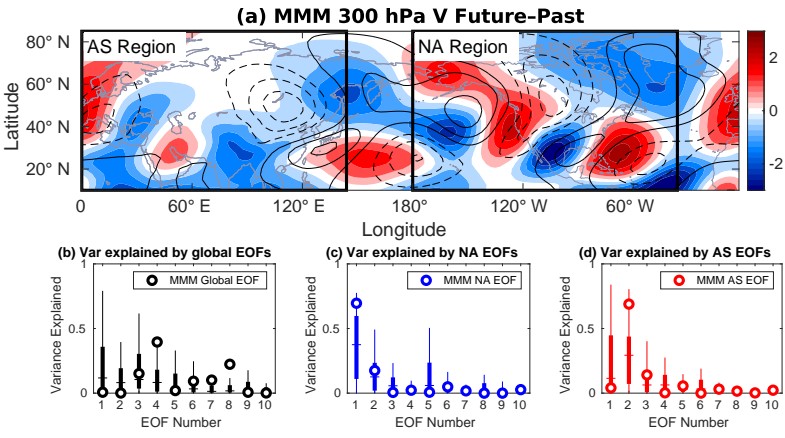

**Figure 4.** a) Multi-model mean 300 hPa DJF meridional wind. Contours show Past (1979-2005) climatology, and shading denotes the anomaly between Future period (2070-2099) and Past; Boxed regions are the domains used for regional EOF calculations and projections hereafter (AS - Euroasia; NA - Pacific North America). Contour interval is 3 $ms^{-1}$ and the zero contour is omitted; b-d) Percentage of spatial variance of every model's anomaly (similar to the shading in a), as explained by its leading ten global (b) and regional (c,d) EOFs. Horizontal lines are the ensemble median, bold vertical bars cover the interquartile range, and thin bars mark the top and bottom quartiles. White circles denote the results for the same calculation, performed with the MMM EOFs and anomaly.

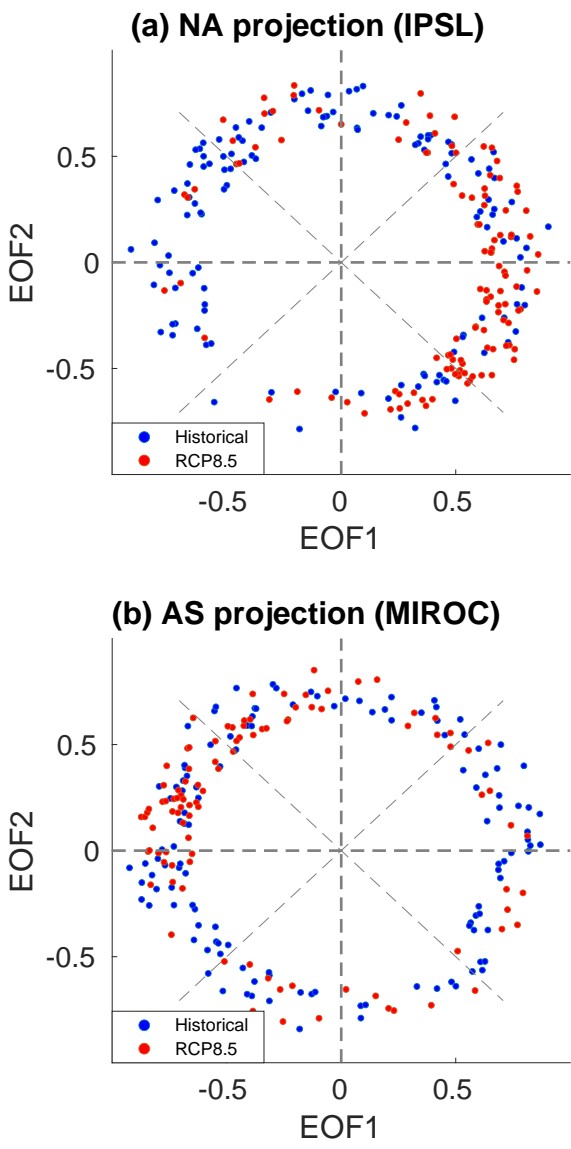

**Figure 5.** Examples of the monthly phase space. Projection of 300 hPa monthly DJF V anomalies onto regional EOFs in two ensemble members: IPSL-CM5A-MR in the NA region (a), and MIROC-ESM-CHEM in the AS region (b). Blue (red) dots are Historical 1900-2005 (RCP8.5 2006-2099) monthly deviations from a 1900-1930 climatology. For clarity, only strongly projected months were included.

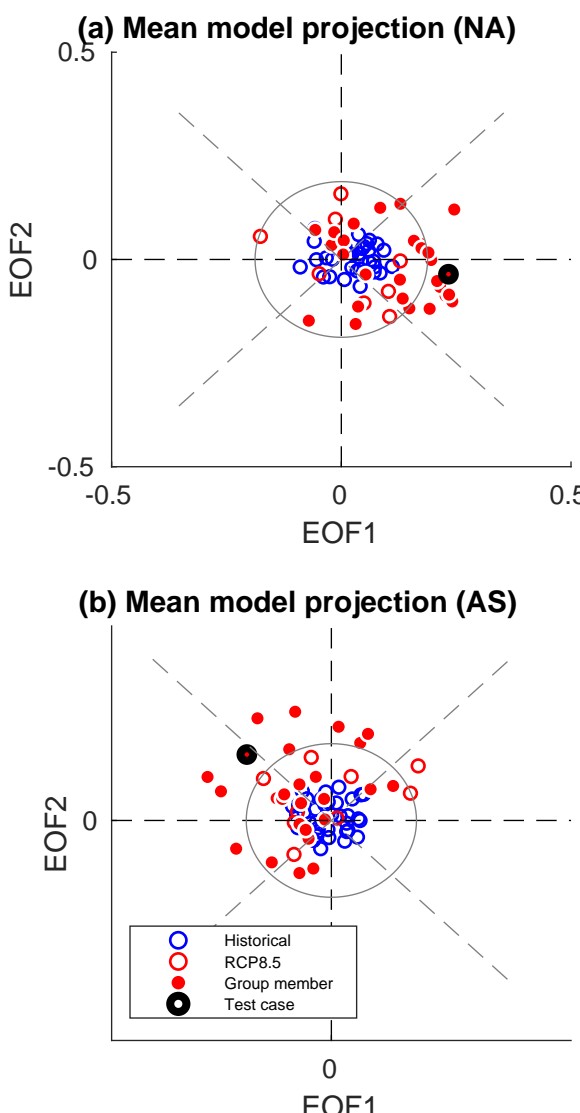

**Figure 6.** Mean location of every model on the (a) $EOF_{NA}$ and (b) $EOF_{AS}$ phase space. Each blue (red) marker represents the mean projection score of all Historical (RCP8.5) DJF months. Filled circles are models that were classified to the preferably-phased groups, with a score bigger than $1\sigma$ (denoted by the grey circle). Highlighted markers show the two GCMs chosen as test cases for daily data projections.

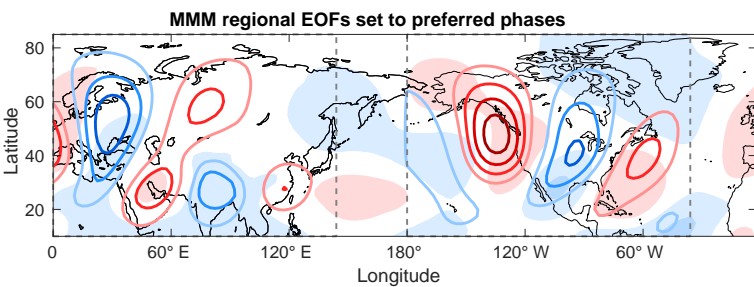

**Figure 7.** Linear combination of regional EOFs. In contours: $\cos(\gamma)V_{EOF1} + \sin(\gamma)V_{EOF2}$. $\gamma$ is set to represent each group's approximate preferred phasing ($-\pi/8$ for NA, $3\pi/4$ for AS). Contour interval is $1\ ms^{-1}$. Light shading is the MMM Future-Past V trend, and dashed lines define the domains of the regional EOFs.

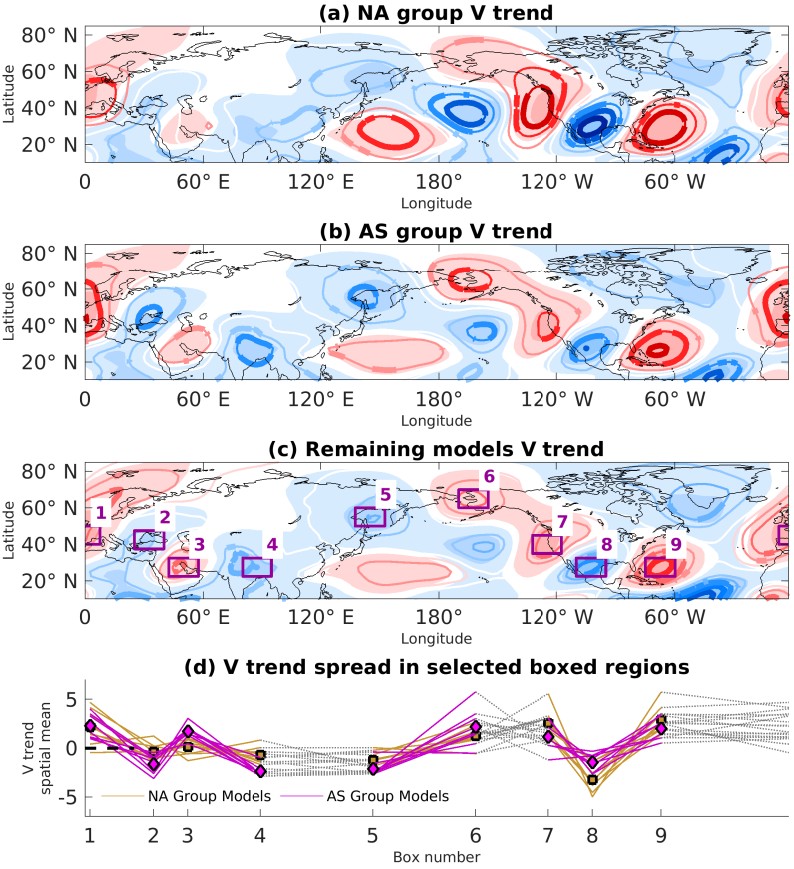

**Figure 8.** The Future-Past 300 hPa DJF V climate trend for all models in the NA (a), AS (b) and NONE groups (9, 8 and 17 models, respectively). The two models with preferred phasing in both regions (BOTH group) were added to the (a) and (b) composites. Contour interval is 1 $ms^{-1}$, and the bold contours signify areas where over 90 % of group members share the same sign with the composite. In shading, the MMM V trend as in Figure 4a. d) shows the spatially averaged V trend ($ms^{-1}$) in the nine boxed regions in (c), for every model in the NA and AS groups (orange and magenta, respectively). Square (diamond) markers denote the NA (AS) group mean for every box. Some lines are partially grayed out to better indicate separate geographical regions.

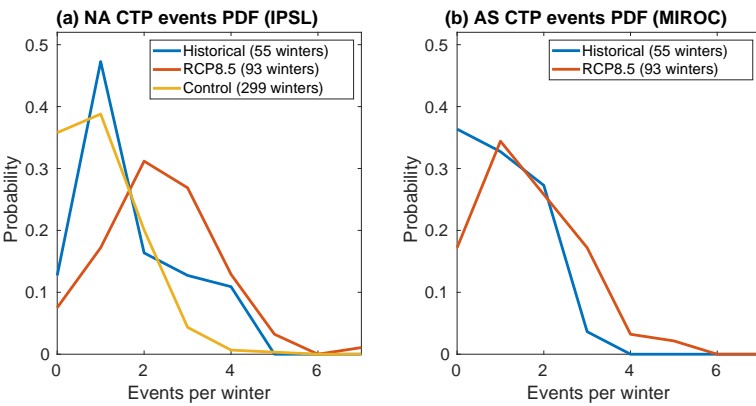

**Figure 9.** Histograms of the probability of CTP event frequency per winter, for (a) IPSL-CM5A-MR and the NA region and (b) MIROC-ESM-CHEM and the AS region. The blue, red and yellow lines signify the Historical (1950-2005), RCP8.5 (2005-2099), and Pre-industrial Control (1800-2099) runs, respectively.

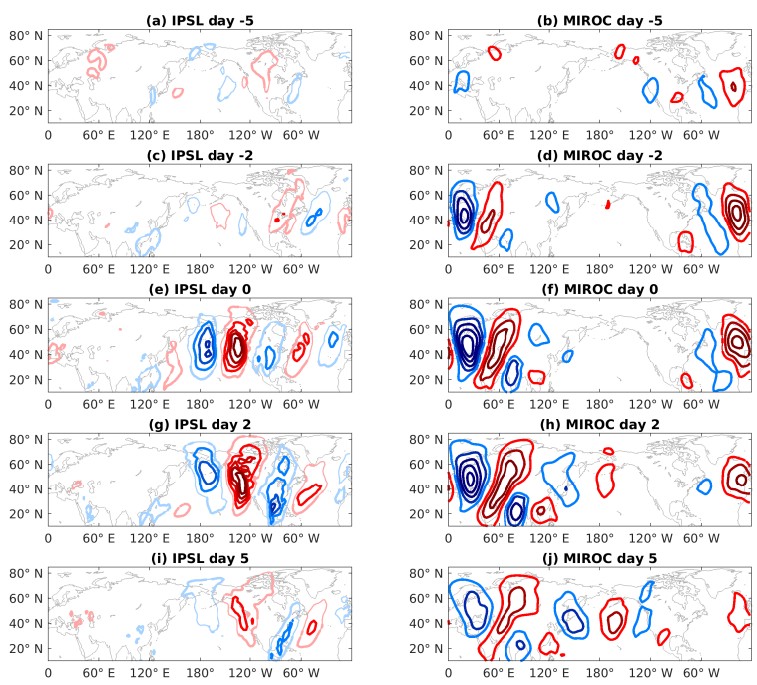

**Figure 10.** Lag composite of CTP events for days -5, -2, 0, 2 and 5. The composites are comprised of 500 hPa daily V deviations from RCP8.5 mean (2006-2099), for IPSL-CM5A-MR (a,c,e,g,i) and MIROC-ESM-CHEM (b,d,f,h,j). The IPSL (MIROC) composite consists of 222 (165) events. Contour interval is $1\ ms^{-1}$ for the IPSL plots, and $2\ ms^{-1}$ for the MIROC plots. Thick contours mark 95 % statistical confidence under a bootstrap two-tail t-test.

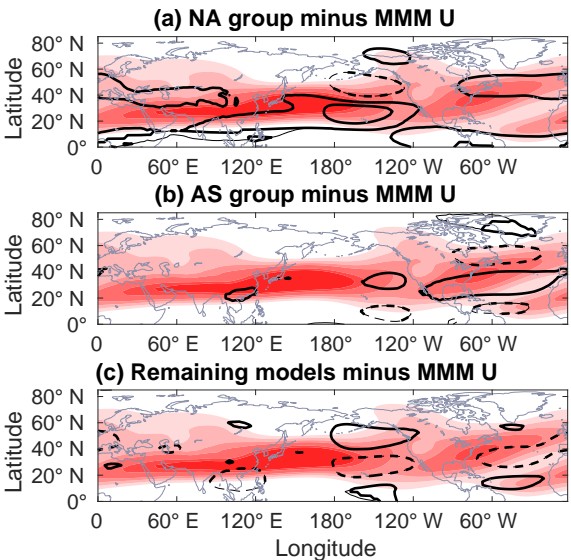

**Figure 11.** Jet stream spread in the ensemble. Data shown is monthly 300 hPa DJF zonal wind group composites for the Past period (1970-2005), with the MMM pattern removed. Contour interval is $1\ ms^{-1}$, with negative values denoted by dashed lines. Thick contours show areas where at least 90 % of group members have the same sign as the group mean. Shading represents the MMM 300 hPa climatological jet stream averaged over the same period ($U \geq 10\ ms^{-1}$, shading interval is $5\ ms^{-1}$).

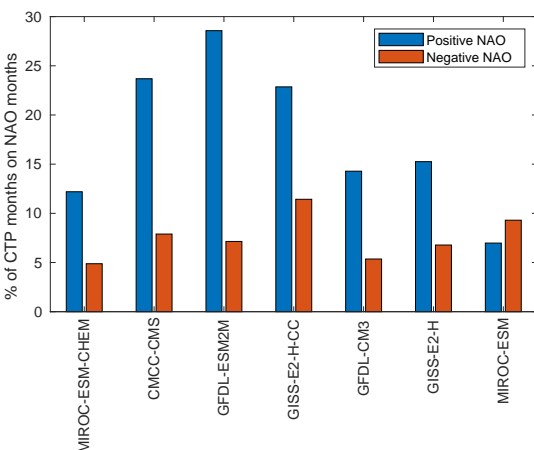

**Figure 12.** Share of preferably phased RCP8.5 CTP months in the AS domain, that occur concurrently with a positive (blue) or negative (red) NAO phase. All models shown are from the AS group.

**Table 1.** CMIP5 model groups based on preferred regional CTP phasing of monthly mean data projection. Models where global EOF2 and 3 are well-separated are denoted by (*). When all three leading functions are well-separated, the models are denoted by (**).

| NA Group | AS Group | BOTH Group | NONE Group | |
|---|---|---|---|---|
| bcc-csm1-1 | CMCC-CMS | IPSL-CM5A-LR** | ACCESS1-0 | GISS-E2-R-CC** |
| bcc-csm1-1-m** | GFDL-CM3* | CSIRO-Mk3-6-0** | ACCESS1-3** | HadGEM2-CC** |
| CESM1-BGC | GFDL-ESM2M* | | CCSM4** | HadGEM2-ES** |
| CMCC-CESM* | GISS-E2-H* | | CESM1-CAM5** | MIROC5 |
| CMCC-CM** | GISS-E2-H-CC | | CESM1-WACCM** | MPI-ESM-LR** |
| HadGEM2-AO** | MIROC-ESM** | | CNRM-CM5** | MPI-ESM-MR** |
| inmcm4* | MIROC-ESM-CHEM** | | FGOALS-g2** | MRI-CGCM3 |
| IPSL-CM5A-MR** | NorESM1-ME* | | FIO-ESM* | NorESM1-M |
| IPSL-CM5B-LR** | | | GFDL-ESM2G* | |