# Peer review of "Future Wintertime Meridional Wind Trends Through the Lens of"

_Weather and Climate Dynamics, 2020_

## Referee Comment (RC1) · Anonymous Referee #1 · 19 Mar 2020

GENERAL COMMENTS:

The authors present a nice study about the representation and future trends of CTP in CMIP5 models and observations, which is a very important topic to understand circulation patterns in a future climate under global warming. In general the plots are clear and well chosen to support their results. However, it was not always easy to follow their conclusions and some further explanation or proof for some of their statements seems necessary. I would therefore recommend major revision to tackle this before publication.

- the authors put a lot of emphasis in the conclusion on the necessity to better understand the connection between mean flow and associated teleconnection patterns. But in their results the authors mainly discuss wave patterns. Maybe it could be helpful to

put some more focus into the mean flow for the discussion of their results. E.g. the authors successfully did overlay the MMM V trend and the regional EOFs, but Fig. 11 only shows one contour of the mean flow (20m/s). Therefore the reader is not able to interpret the anomalies, because there is no relation to the spatial variance of the mean flow. Further the authors could focus a little bit more on the role of the mean flow in their discussion. This could maybe also include some large scale global flow anomalies associated with NAO, PNA, ENSO, etc. The anomalous patterns in the jet strength in their Fig. 11 seem to show similarity to such known global pattern indices. Further, there are already some studies looking at the connection between large scale flow patterns and the resulting wave response (or how they are associated, not suggesting a cause-effect relationship). I think this was also done in the reference they cite (Souders et al, 2014), so maybe could be worth including in their discussion (how do their findings of wave packet anomalies relate to the finding here)?

- Authors are analysing wave pattern with non-zonal waveguide. Wouldn't it then not make more sense to use the wind perpendicular to the climatology (as representation of the waveguide)? Wolf and Wirth (2017, Diagnosing the horizontal Propagation of Rossby Wave Packets along the Midlatitude Waveguide) have shown that this does have an important impact on identifying the correct path of propagating waves. Here the authors do focus on stationary or quasi-stationary waves, but as soon as the waveguide has a meridional component, the more physical quantity to measure the wave is the wind perpendicular to its waveguide. What is the authors point on this, do they assume they would see an even clearer signal in their results or do they expect that this does not have a relevant impact on their results because of the large scale of the wave patterns they are investigating?

SPECIFIC COMMENTS

- Introduction ——————————————————————

- p.1, line 21: Why do the authors only highlight "shifts" in the climatological mean flow?

The strength of the mean flow is important as well, isn't it?

- p.2, line 37-40: Could the authors please give some more explanation on this for the reader who is not familar with the details of Branstator 2002? As this point is crucial for the paper, it seems that some more explanation on this is well invested. How crucial is the exact way of calculating the EOFs and how should they be calculated? I assume they are calculated for the whole hemisphere, either northern or southern hemisphere. The EOFs are further calculated for DJF using subseasonal anomalies - does this mean the deviation of the meridional wind from the mean of the specific season or the deviation from all seasons taken together? As there is still a shift of the jet during the season, would using the meridional wind as deviation from a running mean or low pass filtered signal change the resulting EOF signals or are they a very robust signal? Further, the first two EOFs do show a very similar but shifted wavenumber 5 pattern with more or less opposite sign? Otherwise this combination given in Equ. (1) would make no sense as it tries to capture the phase of a wave that can have nonzero phase propagation, right? How many variance is explained by these EOFs and are they well separated from the next order EOFs? -> I think this was explained later in more detail, so maybe refering to this here or including a short descriptions to make the reader more familiar with what this method does capture/what the associated pattern represent.

- p.2, lines 48-50 Maybe this description of RWPs must be reformulated? RWPs are not necessarily restricted to 1 or 2 wavelengths. Further they are not necessarily consisting of pairs of troughs and ridges; if one has identified a RWP consisting of a trough and a ridge which further leads to downstream development than there will appear either a trough or a ridge, but not necessarily a pair of both (or by decay of individual anomalies on upstream side).

- p.2, line 51: How do the authors conclude what the value of the phase speed (near-zero) of the RWPs is that contribute to the CTP?

- p.3, lines 63-64: What is the statmenet here? The CTP is already a current wintertime

feature, so what do the authors mean by "CTP will be easily excitable by future greenhouse gas forcing"? The statement is that the CTP pattern will become stronger? Also following lines, the authors mention the "CTP-like trend". This means a strengthening of the current CTP pattern or what does the term "CTP-like" mean here? A CTP-like trend could also be a weakening of this pattern or not?

- paragraph lines 67-73: this paragraph describes to what the authors previously refered as linear RW theory (p.2, line 35), correct? The increase of jet strength in winter does lead to a shift in zonal wavenumber toward smaller values. What do the authors mean here when they say that this is not necessarily the case in boreal summer? The strength of the jet in boreal summer does not have an impact on the stationary wavenumbers? r is the focus here on the excitation/amplification of wave responses than rather a shift of wavenumbers (what exactly are the authors refering to by using "the response")?

- p.3, lines 77-78: why is it surprising that seasonal and subseasonal variance of V behave differently? If there is a shift in the power spectra of meridional wind towards lower wavenumbers this could probably also mean a shift towards lower phase speeds (?). If there is an increase in more quasi-stationary wave patterns the contribution of the faster propagating signal could decrease (what one would probably expect if one sees more stationary wave patterns or do I confuse sth here?).

- Data and Methods —————————————————————

- p.4, lines 99-100: Doesn't this subtraction of the seasonal mean automatically cause some anomalies by the shift of the jet position. What would happen if one would remove something like a running mean of 90 days, would this have a huge impact on the result? -> I think I mentioned this further above already in a bit more detail.

- Fig. 1, colorbar?

- p.5, paragraph lines 139-148: Again, the climatological background field is evolving

over the period of the winter. What is the reasoning to subtract the full DJF mean instead of the climatology for the individual months. Wouldn't that be more physical with the reasoning given in this paragraph (to get rid of the evolving planetary scale for the anomalies)? Did the author estimate the impact of using monthly climatologies instead of seasonal climatologies?

- p.5, line 148: why are the data with low projection scores excluded and how were these 70% chosen. Do results depend on the percentage?

- p.5, lines 149-50: Could the authors explain why the focus should be on models with a specific phasing, which do represent a higher score? What would happen if specific models would produce a persistent pattern of an overall slowly evolving pattern. Let's say often a very strong Ridge over the Rockies with downstream amplified wave pattern, as given by EOF1 in Fig.2. If this pattern would break down towards the mid/end of the season with a more zonal flow, this more zonal flow would appear as opposite sign of the EOF1 (ridge less strong than for climatology, downstream meridional tilt of the jet less strong), wouldn't it? This process would represent the recurrent build up and destruction of a specific wave pattern. If such a case would happen it would not be identified in the average score (it would average to zero), right? Could the author comment on this, why they think something like this is not expected to occur or if it would occur, why it would be fine if the score would still identify it as close to origin (not specific relevant wave pattern case)? Or differently formulated, why is the strong focus exclusively on a prefered phase?

- p.6, line 157 (CTP events): So, usually if the authors talk about CTP they have in mind the patterns derived from the EOF of monthly fields. How does such a CTP relate to the CTP events mentioned here in this paragraph. Do the CTP events using daily data (with a high index) mainly represent two RWPs which are located in a way that they have at the same time the correct phasing relative to the "monthly" CTP pattern? Or are those events only restricted to specific regions (the 144 lon mentioned earlier) not for the whole hemisphere. An eastward propagating (phase propagation) RWP will then be

captured several times moving through this region (after moving one wavelength). This means, if I understand it correctly, two RWPs at lag 0 in the composite, could actually be the same RWP at lag 0 and then something like 5 days later after shifting a whole wavelength further eastward if it consists of more than one wavelength (because in the time in between it will not fullfill the phase criteria given in line 161). Is that correct? What would this mean for interpreting these lag-composites?

- p.6, line 161: This phi_d and phi_m refer to the phase as used for the phase in Equ. (1) which was given by gamma, right? So to be clearer and reduce confusion, it could help to be consistent and name them the same way (gamma_d and gamma_m).

- p.6, line 163: What is the future climatology? is this a similar 30 year average (which period?) as for the historical climatology (1900-1930)?

- Results ─────────────────────────────────

p.6, lines 175-178: How do the percentages compare to the findings of B02, weren't they significantly higher and if so, how can the differences be explained. For the separation of the EOFs it does not really matter if the first two are clearly separated, if they are both used or combined in the index of Equ. (1). However, doesn't the close connection to the third EOF mean that it is difficult to analyse them seperately? How does this change for a longer dataset (NCEP-I), are they well seperated from each other or to the third EOF? What is the consequence for this (as the authors mention specifically this test)?

- p.6, line 185 (Fig.S1 - Fig.3): the multimodel mean has higher absolute values than the individual models. It seems like the Fig. S1 only show the first two contours, ommitting the following ones.

- p.7, line 195: I have some difficulty to interpret those number of 0.78 and 0.5 - what is the score range? 1 is as mentioned a perfect copy. What is the lowest score, 0 or -1? And the lowest score is then the exact opposite of the signal? If that is the

case and looking at Fig. S1, this suggests -1 is the lowest skill score, otherwise 0.5 would somehow be like a wavenumber 5 signal with random phasing compared to observations (which obviously is not the case according to Fig. 3). Probably it would be helpful if the authors could give the reader some idea what this score does tell (apart from the upper boundary).

- p.7, line 1999-200: The quasi-stationary wavenumber 5 pattern is per se not a response of GHG forcing as it is already present in the historical data. Is this comment about the trend and differences of this pattern to the historical pattern? Maybe the authors could clarify this.

- p.7, line 204 (Fig. 4): It could be helpful for the reader if the authors would include here the underlying past climatology. Does the MMM response really shows a convincing wavenumber 5 signal? There seems to be a low wavenumber signal over eastern asia/western pacific and a higher wavenumber response over North America/Atlantic. A Fourier analysis would probably not show a single peak at wavenumber 5. The authors further say that although this wavenumber 5 response shows up in the MMM (Fig. 4a), the individual models trends represent only low scores if projected onto the EOFs. This seems to suggest that the MMM response has a high score, but is this really the case? Looking at the response wave signal path from North America southward into the Atantic, this does not seem to have a counterpart in the EOFs from Fig. 2. Does the MMM shows a much higher score? Further, I like that the authors included the supplementary Fig. S1 showing all models. But I would have been even more interested in the same kind of Figure, but based on future data as there is a very good agreement between the historical EOFs between models and observations, but there seems to be some stronger discrepancies in the wave response for future projections. So it is not really obvious that the future projections go for a CTP pattern or rather increase in the evolution of separated wave responses as can be seen to some extent already in Fig. S1 for some models in the historical data (EOF2 showing a seperate NA wave pattern?).

- p.7, line 207: what exactly is the point that should be tested further? Is it the low projection score?

- p.7, lines 213-214: Why does the division into different regions allow to inspect CTP-like responses? Could it not be that doing the calculations for different non-global regions highlights regional wave patterns that are not circumglobal? Fig. 4b shows that the variance of most models cannot be explained captured by the MMM global EOFs, but mainly by EOF 4 (is that correct interpretation?). Wouldn't it be interesting to show this EOF 4? Is it a non global wave pattern of a higher wavenumber? If so, wouldn't that mean that for the future trend the CTP pattern is less relevant and the projections tend to prefer another global wave response? Why isn't that the interpretation of Fig. 4aa and b.

- p.7, line 217 (Fig. S2): What are the values for shading and contours? Shouldn't be the shading in Fig. S2 be identical to the contours in Fig. 3? But blue values increase in Fig. S2 while at the same time red values decrease in spatial extent. Does this mean that the colour contouring is asymmetric between negative nad positive values in Fig. S2?

- p.7, lines 217-219: This is not shown anywhere or is it? The reader hasn't seen the regional EOFs for the obersvational data. So maybe indicating this by sth like "(not shown)".

- p.7/8, lines 220-224: Any conclusion or interpretation the authors could provide the readers? Comparing the regional to the global EOFs in Fig. S2 I'm surprised this makes so much of a difference. But the wave pattern for the EOF1 in NA (Fig. S2a) seems to show a wave pattern coming from the south and pointing to the south, suggesting that this wave pattern does not feature a CTP pattern. Could the authors provide some explanation and interpretation to their results. It seems here that the authors now just use the regional EOFs because they better represent the trend of the models. But why is that, is there some possible explanations behind these signals or

is this just a useful thing to do to capture the trend, because going to smaller scales usually should allow one at some scale to capture all globaltrends if added together.

- p.8, lines 234-235: I really like the visualization of the data in this phase space diagram of Fig. 5. However, what is the measure of uniformly spread? Starting to look at Fig. 5a this seems a bit strange, as the negative phase of EOF2 does not seem to occur very often for weak EOF1 values, in one of the eight parts there are only 4 blue dots, whereas in others there are more then 20. So it is not really uniformly. Could the authors quantify this to some extent with sth like X % in a 90 degree angle range? That shouldn't be too difficult and this would allow them to make their point more convincingly of how much more the RCP8.5 data is concentrated into a specific region of the phase space diagram. Probably not necessary, as now I realize this is what Fig. 6 does. Further to this, Fig. 5 does only show one model, right? This should be mentioned by the authors (it is described as all models in the text), so I totally missed this at the beginning.

- p.8, line 238: Is it obvious that models shows a preference for either NA or AS regions. Is it necessarily "either" for being able to explain why they not project strongly on the global pattern? Isn't it possible that one model prefer boths, some periods with increased wave patterns over the AS which ressemble the given EOFs there, and some periods with increased wave patterns in the NA region with some arbitrary signal everywhere else. The authors seem to exclude this possibility. What is the reasoning for their conclusion?

- p.8, line 244 (Fig. 6): How is the circle calculated? Not clear to me where the assymetry of the circle (right vs left and top vs bottom of the phase space) comes from. This is the standard deviation in physcial space of the individual model monthly data also projected onto the EOFs and then shown in this phase space? If it is the standard deviation of the values in this phase space (as it does sound like in the text), shouldn't it have an equal distant from the origin - or does the circle only seems asymmetrical because of the dashed lines which do not represent an adequate measure of distance?

Maybe use the same order (NA on top of AS) as in Fig. 5. I made this assumption switching from Fig. 5 to Fig. 6, which did confuse me for a moment.

- p.8, lines 246-247: I'm still confused with everything related to this sigma-measure. Aren't most of the rectangles of RCP8.5 inside the circle? The filled ones represent the same models (the group) in Fig. 6a and 6b, right? But this would mean that most models don't get above 1 sigma for any region or do I misunderstand the plots? Further, isn't the standard deviation calculated from the RCP8.5, so wouldn't we expect most models to be inside of the circle by definition (for simplicity assuming a normal distribution)?

- p.8, lines 247-249: Maybe the authors could spend some more time better explain this. I was expecting them to refer to the red rectangles (the mean projection of the RCP8.5 models), but I cannot identify those numbers? First of all, I assume the authors refer to rectangles outside of the circle, but they mention only the sign of the EOF (which would include all of the red rectangles). Further I can identify the eight rectangles outside of the circle in the NA region but not for the AS region (is that because they are on top of each other?).

- p.8, line 251: Again, do the authors really mean non-zero or outside of the circle, because their statement is only true for those outside of the circle, isn't it? Following line as well, this is about the rectangles outside of the circle. The 7 cases are then the 6 inside the quadrant and the one with nearly 0 EOF1 very close to it, right? But how can these numbers be associated with the previous statement of 8 cases with negative (positive) EOF1 (EOF2) for the AS, (representing the second quadrant)?

- p.10, line 288: How is it filtered? Difficult to follow in detail, if it is just mentioned that the data is filtered without specifying how.

- p.10, lines 290-293: Can the authors say anything about the persistence of the CTP events?

- p.10, lines 296-297: Is this really the case? The regions are very different with a shift of partly 60 degrees and huge overlaps. The NA region here captures main parts of both, the Pacific and Atlantic region in Souders et al. (2014). But I assume it depends for what this comparison is used and there is no need for a good agreement between the chosen regions. The authors directly refer to Souder's result of RWP formation, in which case most of the RWP formed in the Pacific region will result in amplified RWPs over the NA region given here, whereas the formation of RWPs in Souder's Atlantic region will contribute to the RWP in the AS region given here - although a direct comparison is not simple because Pacific RWPs could also go all the way towards the AS region whereas RWPs formed in the Atlantic can also decay in the Atlantic.

- p.10, lines 297-298: Difficult to compare as Souder's climatology also captures faster propagating RWPs with wavenumbers around 10. So I agree with the authors that Souder's climatology should give a high upper boundary. The waves the authors are interested in (low wavenumbers around 5) should be more comparable with the quasi-stationary waves investigated in Wolf et al. (2018, Quasi-stationary waves and their impact on European weather and extreme events). Their Fig. 9 should probably give a lower boundary for the wave occurrences with about 0.6 per season in ERA Interim for very persistent wave signals (minimum 10 day lifetime). So the findings here seem to be well in this range of different climatologies.

- p.10, line 306: Maybe not using the term "wave activity", which would be strictly speaking a measure of wave strength/amplitude, but here the authors rather mean frequency.

- p.10, lines 305-309: The authors are measuring also the persistence of the the signal (as this is part of the definition of a CTP event). Would be nice if they could make a statement about the persistence of those events. This should also be sth of interest in terms of climate extreme, because such persistent signal do often lead to extreme events. So knowing about the trend in the persistence could give some hints about the evolution of possible extremes.

- p.10, lines 312-313: Why is the RWP at its peak of the life cycle at lag 0? Strictly speaking, one cannot tell this with the applied measure which can only tell its projection peak onto the CTP. But even this probably does not occur at lag 0, or does it? Lag zero is defined as the first of at least 3 consecutive days when the projection value excees the given threshold. But then, would it not be more likely that the peak occurs around day 1 or slightly afterwards?

- p.10, lines 315-316: where does this conclusion come from? What characteristics specifically and how where they affirmed to be realistic? Where does the conlusion for near stationarity comes from? Not clear where the conclusion and description of RWP comes from in this paragraph. Could the author give some more details and explanations for this paragraph?

- p.11, lines 320-325: How do the author conclude that the wave source is located over Southeast Asia (IPSL case) when there is no statistical significant signal over southeast asia and no clear wave propagation. Further lag 2 and 5 seem to indicate eastward moving phases, contradicting the assumption of a stationary or near stationary signal. This raises again he question if there are not also RWPs with nonzero phase speed. If that would be true, this would mean that individual RWPs could be accounted for more than once (if they include more than one wavelength) which would further be problematic for the identification of the source of this signal or its overall pattern. Further, how many cases are included in this composite? Shouldn't it be something like 1-3 events per season (mentioned last page) for this 93 year period? It is therefore somehow surprising that there is such a strong distortion of the wave signal at positive lags. Do the authors have an idea where this is coming from? Concerning the amplification of the signal, is that not a result of the composite for which the RWPs are forced to have equal phasing around day zero, but no such constraints exists for the days before and afterwards, why one would expect the signal to smooth out and decrease in strength.

- p.11, lines 340-342: Could the authors explain this more thoroughly how the MM trend (Fig. 4a) can be understood as result of the NA and SA related wave patterns? They

add up together to explain the trend given in Fig. 4a? Maybe the authors give some further explanation for the reader.

- p.11, lines 344-350: What are the implications or the main conclusions here? The main take-away message is that there is a strong jet anomaly upstream of the wave signals? Do the authors have some ideas or hypothesis why the jet should be modified in this way for being able to be associated with a strong wave signal downstream? Nice findings. Do the authors have the information about the connection of their patterns to large scale pattern indices as Fig. 11b looks like a positive NAO (correct?)? Which would be very much in agreement to the findings of Wolf et al (2018, Quasistationary waves and their impact on European weather and extreme events), where they showthat a strong increase in QSW activity along the subtropical jet and the Mediterranean region (EOF2 and 4 in their Fig 11) can be associated with a positive NAO phase. So is the SA group associated with models showing more frequent positive NAO phases? Similar conclusion could be concluded from the PNA, which in its negative phase leads to increased activity of quasi-stationary waves over NA (EOF1 and 3 in Fig 11, wolf et al 2018).

- p.11/12, lines 351-355: This is rather a question, referring again to a point mentioned earlier: are the authors sure about the compositing of RWP events (is there no double counting for lagged RWPs)? Because those double counting with time lags could obscure the temporal relation between tropical forcing and the associated RWP.

- Discussion and Conclusion ———————————

- p.12, lines 369-370: But this was not working for the projection onto the global CTP, was it? This result was associated with the separation of the signal into different regions. So why can the results be associated with the evolution of the overall CTP? It is not obvious why the results for the analysis, restricted to specific regions, should explain the behaviour of a circumglobal teleconnection pattern. Could the authors maybe make this clearer?

- p.13, lines 418-420: This link between mean flow (waveguide) and the resulting wave pattern seems indeed crucial. The authors show the spatial relationship which seems to show increased/shifted jet stength upstream of the onsetting wave pattern. Do the authors have some interpretation or insight into the dynamical link for this connection, or is this just a result of increased jet strength (shifted away from) in the locations were are climatologically seen less activity of occurring wave patterns (for those regional wave groups)?

---

## Referee Comment (RC2) · Anonymous Referee #2 · 30 Mar 2020

Sandler & Harnik analyse the sub-seasonal structure of the wave-5 circumglobal tele-connection pattern in northern hemisphere winter circulation. Using on reanalysis- and model data they identify preferred phases, investigate the models' skill in the representation of the pattern, discussing the regional structures and respective trends.

Major Comments

Are the EOFs for DJF the same as for each month separately? Is there one Month that dominates the seasonal signature? For reference, Ding and Wang 2005 showed that the CTP had different signatures throughout JJA.

Could the authors quantify which models have been most accurate in the representation of the CTP compared to reanalysis? In the light of the 'strong disagreements'

between models, this might allow some careful statements which model is more reliable in terms of future projections.

Could the authors expand on what intrinsic mid-latitudinal mechanisms might trigger and maintain the CTP and its preferred phase?

Some unusual wording and unspecific terminology make it hard to understand the authors at times (see minor comments for some examples).

Minor Comments: -Include 'wintertime' in title. Further 'teleconnections' refer to patterns such as 'ENSO' or 'MJO' could this be further specified in the title to avoid confusion? -(p.1, l.2) '..variability ARE upper tropospheric..' -(p.1,l.4) Others have used the abbreviation CGT (see e.g. Ding & Wang 2005), consider changing CTP to CGT to stay consistent with the terminology used in the literature. -Try to avoid effusive / inessential expressions such as 'dramatically' (p.1. l.1), 'surprisingly' (p.1 l.4), (p.3. l.78) and (p.12 l.381), 'first described two decades ago' (p.2,l.25) 'most definitely' (p.14 l.437), 'unsurprisingly' (p.9 l270), 'most definitely' (p.14, l.437) -Could the authors add a few sentences on differences to summer Circumglobal / stationary waves to the introduction? -(p.1, l.4) Maybe change 'likeliness .. emerges' their 'frequency increases' or similar. -(p.1, l.6) Name the timescales (Monthly and 3-day mean right?) -(p.1,l.11) 'This categorization strongly corresponds to the ensemble spread in local trend magnitude.' It is not clear to me what this means in this context. -(p.1 l.15) 'Thus, we conclude that this hemisphere-wide climate change signature is actually comprised of several regional effects'. –What hemisphere wide climate change signature? Better use 'response'. Also the authors highlight in the paragraph before, that changes are found visible on a more regional level, how is it a hemisphere-wide signal? Please be a bit more concise. (p.1, l.20) 'Projections of future circulation trends, driven by anthropogenic climate change, commonly display large scale patterns.' It feels like this statement requires a reference. (p.1 l.21) '..in order to provide dynamical reasoning and theory.' In order to test hypotheses and theories? (p.1 l22.) '..development on finer scales?' What scales, higher temporal resolution? (p.1 l22.) 'changes in subseasonal to seasonal fluctuations' –

changes in variability or changes in subseasonal circulation patterns? (p.2 l.26.) 'term' change to 'pattern' (p.2 and later) The 'CTP' is described as 'a pattern' and then as 'the wave' or and then as 'waves'. Later it is described as 'a class of related patterns', all of them 'waves'. It would be helpful if the authors could rewrite that part while being more precise in terminology. The sentence in l. 37 should come a bit earlier to clarify the hierarchy among the terms, which are seemingly used synonymous earlier in the paragraph. (p.3. l70) is it an acceleration or a poleward shift (or both)? (p.23, l.74) works -> studies (p.3 l.79) please further specify what the conceptual gap is. (p.4 l.98) do the patterns depend on the chosen mid-lat range? (p.4 l.105) this sentence seems grammatically wrong? (p.5 l. 143) over which years is the climatology defined for re-analysis datasets? (p.5 l. 148 ff) Wouldn't a negative projection score mean a preferred phase opposite to the one in question while a score of zero would refer to an arbitrary phase? (p.6 l.159) why is a running mean of three days chosen? (p.6 l.163) 'future' (p.6 167) OLR – provide full expression before using an Acronym, here: outgoing long-wave radiation? (p.6 l171) What are the signatures of the other EOFs? Are they more local and excluded from the analysis for that reason? (p.7. l. 220) Please be more specific, I don't understand this sentence. (p.9 l.264) future , past (p.9 l.275) what is meant by temporal frequency here, their occurrence on subseasonal timescales? (p.9 l.281) can this statement be quantified? (p.9 l.286) as a three-day running mean was applied it is incorrect to speak of days in this context. Better use 'timestep' or similar. (p.10. l.288) How are events filtered? (p. 10 l. 290) change 'observational' to 'reanalysis' (here and everywhere else) (p.10 l. 299) 'much along the lines' -> similar to (p.11 l.340) The conclusion is hard to understand, could this be re-formulated? In the Discussion / Conclusion section: Could the authors provide the Figures in which each of the discussed findings is shown? (p.12 l.363) change 'business as usual' to 'high emission'. (p.12 l.364) what does 'decent skill' mean in this context? (p.12 l.369) add a short statement on consquences for predictability / future surface weather. (p.12 l.379). Where are those regions? (p.12 l.383) Reference? (p.12 l.385) 'Seemingly'? Does it or doesn't it? (p.13 l.397) 'However...' I don't understand this sentence. (p.13

l.406) 'Another important scale is the spatial one'. Consider removing this sentence. (p.14 l.435) 'There is difficultie in singling out..' Could the authors be more specific? (p.14, l.437) provide reference (p.14 l.441) What else could provide relevant forcing? Consider citing Garfinkel et al. 2020 (https://journals.ametsoc.org/doi/10.1175/JCLI-D-19-0181.1?mobileUi=0) (p.14 l.446) PV = potential vorticity (p.14 l. 447) 'It should be interesting. . .' -> 'Future analysis will focus on. . . '
* * *

---

## Author Comment (AC1) · 3 Apr 2020

Thank you for your thorough and helpful comments. We will incorporate all of your notes in our revised paper, but in the meanwhile we'll use the discussion platform to shed light on some of the major themes in your comment.

The CTP across scales: An important framework for interpreting our results is analyzing the CTP as a unique bridge between different temporal and spatial scales. Some patterns (like the PNA) only dominate seasonal or interannual timescales. The CTP, however, is an accumulation of quasi-stationary synoptic components which are also clearly manifested on subseasonal scales due to their near-zero phase velocity. That is why we define the CTP as a family of related patterns, as Branstator (2002) did in his

original paper. This distinction will be better explained and highlighted in the revised paper. All the patterns in the CTP family are number-5 QSWs produced by combinations of the EOFs (global or regional), with possible different phases and locations. This specific family of QSWs is the focus of our study because of its surprising robustness. These patterns are found through various methods (one-point correlation maps, EOF analysis, objective tracking) in observations and models (from dry primitive equation models to fully coupled GCMs), once one looks at monthly means of subseasonal (seasonal means removed) patterns for the EOFs, or seasonal mean variance for one-point time lagged correlations. We therefore consider the regional non-circumglobal patterns to be local manifestations of the CTP. This can also help explain why projection scores on global EOFs are low. The global EOFs capture a combination of related regional patterns. However, as we learn from the regional analysis, separate CTP sectors can have differently phased local waves, so looking at the entire hemisphere is limiting in terms of degrees of freedom. The RWPs themselves are a product of regression onto regional EOFs. Therefore, one often finds localized wave packets that circumscribe the globe, but that is not always the case. For example, one can think of a specific month with a strong northerly flow over North America, corresponding to a positive EOF1. However, as the wave itself is not circumglobal, the flow over Europe and the Mediterranean might have waves with a different phase or even no wave signature. In that case, the projection will be strong for the regional NA EOF1, but poor for the global EOF1.

The CTP as a climate change signal: The CTP is not a phenomenon that is unique to future climate, but its presence is clearly seen in the projected long-term trend. It is tricky to answer the question of whether the CTP will become stronger as a result of climate change. There isn't an amplified monolithic wave that instantaneously spans the entire hemisphere in the future, but we posit that the projected stronger wavenumber-5 signal (seen in subseasonal and climatological data) is related to a frequent excitation of persistent RWPs with specific phases (seen in daily data). This idea has been previously explored in more theoretical terms. Branstator & Selten (2008) found that GHG

forcing is projected to excite various modes of internal variability more frequently, and among them is the synoptic manifestation of the CTP. On a side note, while it is true the resulting global signature is not a pure number-5 wave throughout, we find it convincingly robust. The Fourier analysis performed in Simpson et al (2016) for this exact trend reveals a significant wavenumber 5 component.

Wave response and the mean flow: Due to its timescale-spanning nature, it is hard to disentangle the effects of the mean flow from the CTP itself. It has been previously shown, as you've mentioned, that jet configuration affects teleconnectivity. The main take-away message from our mean flow analysis is that the local mean flow anomaly that relates to CTP phase preference (upstream narrowing of the jet) isn't an artefact of the CTP analysis, but rather an intrinsic characteristic of the models themselves. This is because this feature is seen in the climatological bias of the models in historical data (compared to the MMM historical climatology). Our working hypothesis is that local jet structures might allow for RWPs with specific phases to have greater persistence and reach. We focus mainly on models with a single phase preference in order to isolate this connection clearly and avoid analyzing multiple (sometimes opposing) patterns. We will try to further demonstrate our results for the mean flow in the revised version. Also, incorporating other oscillations is a good idea (especially NAO which is locally very connected to the CTP; Yuan et al., 2011).

Analysis of wind perpendicular to the waveguide: Our work mostly relies on EOF analysis, which unlike RWP tracking, is not limited to the horizontal propagation of the waves. The meandering shape of the zonal waves in the EOFs shows that they capture the climatology of the waveguide. However, this is not to say that the method shown in Wolf & Wirth (2017) won't be useful. In the daily calculations, using perpendicular flow might help us obtain a slightly cleaner picture (better RWP accounting), but we don't expect it to meaningfully alter our conclusion.

References

Branstator, Grant. "Circumglobal teleconnections, the jet stream waveguide, and the North Atlantic Oscillation." Journal of Climate 15, no. 14 (2002): 1893-1910.

Branstator, Grant, and Haiyan Teng. "Tropospheric waveguide teleconnections and their seasonality." Journal of the Atmospheric Sciences 74, no. 5 (2017): 1513-1532.

Branstator, Grant, and Frank Selten. ""Modes of variability" and climate change." Journal of Climate 22, no. 10 (2009): 2639-2658.

Simpson, Isla R., Richard Seager, Mingfang Ting, and Tiffany A. Shaw. "Causes of change in Northern Hemisphere winter meridional winds and regional hydroclimate." Nature Climate Change 6, no. 1 (2016): 65-70.

Wolf, Gabriel, and Volkmar Wirth. "Diagnosing the horizontal propagation of Rossby wave packets along the midlatitude waveguide." Monthly Weather Review 145, no. 8 (2017): 3247-3264.

Yuan, Jiacan, Steven B. Feldstein, Sukyoung Lee, and Benkui Tan. "The relationship between the North Atlantic jet and tropical convection over the Indian and western Pacific Oceans." Journal of climate 24, no. 23 (2011): 6100-6113.

---

## Editor Comment (EC1) · Tim Woollings (Editor) · 6 Apr 2020

I would like to thank the reviewers for their constructive comments, and the authors for the interesting discussion. The reviewers have suggested some areas of the manuscript which would benefit from additional explanation and information. I look forward to reading a revised version of the paper addressing these areas. Best wishes Tim Woollings
* * *

---

## Author Response (AR1)

**Point by Point Response & Changes to Manuscript**

R – Referee comment ; A – Author response ; C – Manuscript change

**Referee #1:**

**Major Comments:**

**R**: the authors put a lot of emphasis in the conclusion on the necessity to better understand the connection between mean flow and associated teleconnection patterns. But in their results the authors mainly discuss wave patterns. Maybe it could be helpful to put some more focus into the mean flow for the discussion of their results. E.g. the authors successfully did overlay the MMM V trend and the regional EOFs, but Fig. 11 only shows one contour of the mean flow (20m/s). Therefore the reader is not able to interpret the anomalies, because there is no relation to the spatial variance of the mean flow. Further the authors could focus a little bit more on the role of the mean flow in their discussion. This could maybe also include some large scale global flow anomalies associated with NAO, PNA, ENSO, etc. The anomalous patterns in the jet strength in their Fig. 11 seem to show similarity to such known global pattern indices. Further, there are already some studies looking at the connection between large scale flow patterns and the resulting wave response (or how they are associated, not suggesting a cause-effect relationship). I think this was also done in the reference they cite (Souders et al, 2014), so maybe could be worth including in their discussion (how do their findings of wave packet anomalies relate to the finding here)?

**A:** We added a more detailed plot showing the jet bias, while highlighting the importance of it being found in the Historical model bias (meaning that this is not an artefact of the transient CTP pattern itself). We also added an explanation into a possible link between the CTP and NAO. Using Wolf et al (2018) and Watanabe (2004), we found that in the ensemble, the AS pattern is much more likely to occur during the positive NAO phase. This means that future NAO trends (specifically, a positive shift) might force more specific CTP phases. This is now detailed throughout the article, and shown in Figure 12. As for the PNA, unlike the NAO, we find it hard to separate this mode from the synoptic scale NA RWPs, as they share roughly the same domain.

**C**: Background - Specifically, the NAO can act as a precursor to such RWPs, affecting their path and amplitude (Watanabe, 2004; Wolf et al., 2018).

Methods - To test the NAO's possible role in forcing the CTP, we use a standard NAO index (Hurrell et al, 2003): projection of monthly sea-level pressure (SLP) anomalies onto the leading EOF of the seasonal SLP anomaly in the North Atlantic sector (20° N-70° N; 90° W-40° E). Positive and negative NAO phases are defined as months when the index exceeds one standard deviation (+-sigma).

Results - As for other large scale modes which can trigger the CTP, the phase of the NAO was found to be correlated with the occurrence of monthly CTP preferred patterns in the AS region. For seven models of the AS group, we calculated the share of preferably phased Future CTP months to happen concurrently with a positive or negative NAO phase. For six of the seven models, the monthly AS CTP pattern is at least twice as likely to occur with a positive NAO phase (Fig. 12). No such relation was found for preferred NA phases (not shown).

Discussion - Large scale internal modes also act as forcing for the CTP. The monthly AS pattern, specifically, was found to be correlated with phasesof the NAO, being twice as likely to occur during the positive NAO phase compared to the negative one. This connection is in agreement with previous studies dealing with quasi-stationary wavepackets and the NAO. Wolf et al (2018) found a strong correlation between negative seasonal NAO values and decreased wave activity in the midlatitudes. Additionally, RWPs reminiscent of the preferred AS pattern can be found in Watanabe (2004), where they are identified as an Asian extension lagging positive NAO events. It is therefore

possible that, within the AS group, some common trend for NAO variability (like a general shift towards its positive phase) forces this regional CTP pattern. This hypothesis is beyond the scope of this study.

R: Authors are analysing wave pattern with non-zonal waveguide. Wouldn't it then not make more sense to use the wind perpendicular to the climatology (as representation of the waveguide)? Wolf and Wirth (2017, Diagnosing the horizontal Propagation of Rossby Wave Packets along the Midlatitude Waveguide) have shown that this does have an important impact on identifying the correct path of propagating waves. Here the authors do focus on stationary or quasi-stationary waves, but as soon as the waveguide has a meridional component, the more physical quantity to measure the wave is the wind perpendicular to its waveguide. What is the authors point on this, do they assume they would see an even clearer signal in their results or do they expect that this does not have a relevant impact on their results because of the large scale of the wave patterns they are investigating?

A: Our work mostly relies on EOF analysis, which unlike RWP tracking, is not limited to the horizontal propagation of the waves. The meandering shape of the zonal waves in the EOFs shows that they capture the climatology of the waveguide. However, this is not to say that the method shown in Wolf & Wirth (2017) won't be useful. In the daily calculations, using perpendicular flow might help us obtain a slightly cleaner picture (better RWP accounting), but we don't expect it to meaningfully alter our conclusion.

**Minor Comments:**

**R:** p.1, line 21: Why do the authors only highlight "shifts" in the climatological mean flow? The strength of the mean flow is important as well, isn't it?

A: Changed "shifts" to "changes" to avoid confusion.

**R:** p.2, line 37-40: Could the authors please give some more explanation on this for the reader who is not familar with the details of Branstator 2002? As this point is crucial for the paper, it seems that some more explanation on this is well invested. How crucial is the exact way of calculating the EOFs and how should they be calculated? I assume they are calculated for the whole hemisphere, either northern or southern hemisphere. The EOFs are further calculated for DJF using subseasonal anomalies - does this mean the deviation of the meridional wind from the mean of the specific season or the deviation from all seasons taken together? As there is still a shift of the jet during the season, would using the meridional wind as deviation from a running mean or low pass filtered signal change the resulting EOF signals or are they a very robust signal? Further, the first two EOFs do show a very similar but shifted wavenumber 5 pattern with more or less opposite sign? Otherwise this combination given in Equ. (1) would make no sense as it tries to capture the phase of a wave that can have nonzero phase propagation, right? How many variance is explained by these EOFs and are they well separated from the next order EOFs? -> I think this was explained later in more detail, so maybe refering to this here or including a short descriptions to make the reader more familiar with what this method does capture/what the associated pattern represent.

**A:** Provided further explanation on Branstator's original method while also highlighting the robustness of the patterns. Additionally, we now mention that EOF details are provided later in the paper.

**C:** One method commonly used for isolating these patterns is Empirical Orthogonal Functions (EOF) analysis. In B02, the author calculated the two leading EOFs of the subseasonal anomalies of monthly 300 hPa meridional wind for the entire NH. This means that each season's mean was removed from every monthly mean \$V\$ map. The resulting patterns, a pair of rather similar number-5 waves in quadrature, were found to be very robust. B02 showed that they can be produced from

different fields' EOFs, as well as other methods altogether (like one-point correlation). Further explanations on the EOFs used in this study are provided in Sections 2 and 3.1.

**R:** p.2, lines 48-50 Maybe this description of RWPs must be reformulated? RWPs are not necessarily restricted to 1 or 2 wavelengths. Further they are not necessarily consisting of pairs of troughs and ridges; if one has identified a RWP consisting of a trough and a ridge which further leads to downstream development than there will appear either a trough or a ridge, but not necessarily a pair of both (or by decay of individual anomalieson upstream side).

**A:** Changed the wording so it's clear we're talking about RWPs that are associated with the CTP specifically. Also, removed the mention of trough-ridge "pairs".

**C:** These RWPs, sometimes called wave trains, are local synoptic structures. The ones that are associated with the CTP are typically comprised of one or two wavelengths, seen as a sequence of troughs and ridges in geopotential height, or as alternating northward-southward anomalies in meridional wind. One noteworthy feature of these specific RWPs is their velocity. They have positive zonal group velocity and near-zero phase velocity ...

**R:** How do the authors conclude what the value of the phase speed (nearzero) of the RWPs is that contribute to the CTP?

**A:** This is a common interpretation of the CTP (see references in lines 69-70) – a subseasonal signal which is actually composed of persistent quasi-stationary patterns in the synoptic scale. If the waves had a non-zero phase speed, they would not appear so clearly on subseasonal data.

**C:** This makes the CTP a unique bridge between timescales. The prevalence and quasi-stationary nature of these specific synoptic RWPs allow their influence to manifest on a subseasonal scale as well. ...

**R:** p.3, lines 63-64: What is the statmenet here? The CTP is already a current wintertime feature, so what do the authors mean by "CTP will be easily excitable by future greenhouse gas forcing"? The statement is that the CTP pattern will become stronger? Also following lines, the authors mention the "CTP-like trend". This means a strengthening of the current CTP pattern or what does the term "CTP-like" mean here? A CTP-like trend could also be a weakening of this pattern or not?

**A:** We changed the phrasing so it's clearer that the CTP isn't a future-climate pattern, but rather, its presence is clearly seen in the projected long-term trend.

**C**: It has been previously posited that the CTP, as a low frequency internal mode, will be excited more frequently in the future, as greenhouse gas (GHG) forcing grows stronger.

**R**: paragraph lines 67-73: this paragraph describes to what the authors previously refered as linear RW theory (p.2, line 35), correct? The increase of jet strength in winter does lead to a shift in zonal wavenumber toward smaller values. What do the authors mean here when they say that this is not necessarily the case in boreal summer? The strength of the jet in boreal summer does not have an impact on the stationary wavenumbers? r is the focus here on the excitation/amplification of wave responses than rather a shift of wavenumbers (what exactly are the authors refering to by using "the response")?

**A:** The wintertime CTP trend is mainly explained through linear theory, yes. We expanded a bit on the topic of the boreal summer CTP response, where soil moisture and convective heating play an important role in teleconnection amplification. On that note, "teleconnection amplification" can mean either wavenumber shift or wave amplification, as lo(convective heating, land interactions)ng as the result is a stronger signature of the teleconnection on longer timescales.

**C:** For the summertime waveguide, the driving mechanism might be more complex, as signs show that changes in the diabatic forcing (convective heating, land interactions) play an important role in the amplification of the teleconnection Ten & Branstator, 2019).

**R:** p.3, lines 77-78: why is it surprising that seasonal and subseasonal variance of V behave differently? If there is a shift in the power spectra of meridional wind towards lower wavenumbers this could probably also mean a shift towards lower phase speeds (?). If there is an increase in more quasi-stationary wave patterns the contribution of the faster propagating signal could decrease (what one would probably expect if one sees more stationary wave patterns or do I confuse sth here?).

**A:** We find the Wills et al results surprising because they are specifically analyzing QSW variance on a subseasonal scale (we put more emphasis on this in this version). This means that it's not a decrease in the contribution of fast propagating waves, but rather stationary components. We address this discrepancy further in the Conclusions section.

**C**: on the subseasonal scale, the variance of quasi-stationary waves (calculated using 300 hPa zonal V anomalies) is surprisingly projected to decrease, in an apparent contrast to the seasonal signature

**R**: p.4, lines 99-100: Doesn't this subtraction of the seasonal mean automatically cause some anomalies by the shift of the jet position. What would happen if one would remove something like a running mean of 90 days, would this have a huge impact on the result? -> I think I mentioned this further above already in a bit more detail.

**A:** We recalculated the EOFs using 90 day means for reanalysis and CMIP5, and it does not seem to change our results. For a few models, the patterns in the EOFs are clearer over the AS sector, but for reanalysis and for most of the ensemble there is no major difference between the resulting patterns. When the original EOF is projected onto the running-mean EOF, the majority of models score 0.85 or higher.

R: No colorbar in Fig 1

A: Added a colorbar

**R:** p.5, paragraph lines 139-148: Again, the climatological background field is evolving over the period of the winter. What is the reasoning to subtract the full DJF mean instead of the climatology for the individual months. Wouldn't that be more physical with the reasoning given in this paragraph (to get rid of the evolving planetary scale for the anomalies)? Did the author estimate the impact of using monthly climatologies instead of seasonal climatologies?

**A:** We performed the calculations again with monthly climatology and it did not affect our results. Specifically, changes in the mean projection index (used to determine the model subgroup and preferred phase) were in the order of  $10^{-3}$ .

**R:** p.5, line 148: why are the data with low projection scores excluded and how were these 70% chosen. Do results depend on the percentage?

**A:** Weakly projected months will form an angle on the phase space, but this value doesn't have any physical meaning

**C**: We average all highly projected monthly index values to obtain the mean projection score of every experimental run. This is done after excluding data with low projections scores (below the 70th percentile of index magnitude), as these months' phase index doesn't have physical meaning. Our results are not sensitive to the choice of percentile.

**R:** p.5, lines 149-50: Could the authors explain why the focus should be on models with a specific phasing, which do represent a higher score? What would happen if specific models would produce a persistent pattern of an overall slowly evolving pattern. Let's say often a very strong Ridge over the Rockies with downstream amplified wave pattern, as given by EOF1 in Fig.2. If this pattern would break down towards the mid/end of the season with a more zonal flow, this more zonal flow would appear as opposite sign of the EOF1 (ridge less strong than for climatology, downstream meridional tilt of the jet less strong), wouldn't it? This process would represent the recurrent build up and destruction of a specific wave pattern. If such a case would happen it would not be identified in the

average score (it would average to zero), right? Could the author comment on this, why they think something like this is not expected to occur or if it would occur, why it would be fine if the score would still identify it as close to origin (not specific relevant wave pattern case)? Or differently formulated, why is the strong focus exclusively on a prefered phase?

**A:** Our method only captured wavy months. For a near-zero mean index, a model should have wavy months with different longitudinal phases (for instance positive/negative EOF1 patterns). In the scenario described in the comment, the destruction of the wave pattern (zonal flow) will have a low projection score (below 70th percentile) and therefore will be filtered out.

**R:** p.6, line 157 (CTP events): So, usually if the authors talk about CTP they have in mind the patterns derived from the EOF of monthly fields. How does such a CTP relate to the CTP events mentioned here in this paragraph. Do the CTP events using daily data (with a high index) mainly represent two RWPs which are located in a way that they have at the same time the correct phasing relative to the "monthly" CTP pattern? Or are those events only restricted to specific regions (the 144 lon mentioned earlier) not for the whole hemisphere. An eastward propagating (phase propagation) RWP will then be captured several times moving through this region (after moving one wavelength). This means, if I understand it correctly, two RWPs at lag 0 in the composite, could actually be the same RWP at lag 0 and then something like 5 days later after shifting a whole wavelength further eastward if it consists of more than one wavelength (because in the time in between it will not fullfill the phase criteria given in line 161). Is that correct? What would this mean for interpreting these lag-composites?

**A:** CTP events were defined in order to examine the preferably-phased monthly pattern on a finer temporal resolution. They have the same specific regions and phases. By definition, their projection index (and phase) is persistent for a few days, so we capture only QSWs. That is why waves with phase propagation will not be captured. We changed the wording in the paragraph to clarify these points.

**C:** We define "CTP events" in daily mean data, which are essentially a synoptic manifestation of the wave patterns that are found in monthly data. We capture quasi-stationary Rossby wave packets that are in the same domain and nearly in phase with the preferably phased monthly patterns (only on the 500 hPa level, as explained previously).

**R**: p.6, line 161: This phi\_d and phi\_m refer to the phase as used for the phase in Equ. (1) which was given by gamma, right? So to be clearer and reduce confusion, it could help to be consistent and name them the same way (gamma\_d and gamma\_m).

A: True. Changed to gamma.

**R:** p.6, line 163: What is the future climatology? is this a similar 30 year average (which period?) as for the historical climatology (1900-1930)?

A: 2070-2099, the same period used for the trend (Future minus Past). Added the years to this paragraph

**R**: p.6, lines 175-178: How do the percentages compare to the findings of B02, weren't they significantly higher and if so, how can the differences be explained. For the separation of the EOFs it does not really matter if the first two are clearly separated, if they are both used or combined in the index of Equ. (1). However, doesn't the close connection to the third EOF mean that it is difficult to analyse them seperately? How does this change for a longer dataset (NCEP-I), are they well seperated from each other or to the third EOF? What is the consequence for this (as the authors mention specifically this test)?

**A:** In B02, the global vEOFs explain 22% & 13%, but they are calculated for CCSM3 which tends to overshoot compared to reanalysis. In our analysis, for instance, CCSM4 gets 16% and 13% compared to NCEP's 13.6% and 10.8%. As for the separation, according to North et al, when two functions are not separated it means that their individual patterns are actually a random mixture of the two true eigenvectors. This muddles the physical interpretation of our 2D index, so we tried to avoid it by using NCEP (where the first two functions are separated from the third).

R: p.6, line 185 (Fig.S1 - Fig.3): the multimodel mean has higher absolute values than the individual models. It seems like the Fig. S1 only show the first two contours, ommitting the following ones.
A: The third contour and onward were omitted for clarity (Fig S1 is very dense). We now specify this in the caption.

**R:** p.7, line 195: I have some difficulty to interpret those number of 0.78 and 0.5 – what is the score range? 1 is as mentioned a perfect copy. What is the lowest score, 0 or -1? And the lowest score is then the exact opposite of the signal? If that is the case and looking at Fig. S1, this suggests -1 is the lowest skill score, otherwise 0.5 would somehow be like a wavenumber 5 signal with random phasing compared to observations (which obviously is not the case according to Fig. 3). Probably it would be helpful if the authors could give the reader some idea what this score does tell (apart from the upper boundary).

**A:** Added an explanation for the scale of the index.

**C:** Zonal number 5 patterns will have a large absolute score (1 for a perfect copy, -1 for the same wave in antiphase), and a zero score represents some very dissimilar pattern.

**R:** p.7, line 1999-200: The quasi-stationary wavenumber 5 pattern is per se not a response of GHG forcing as it is already present in the historical data. Is this comment about the trend and differences of this pattern to the historical pattern? Maybe the authors could clarify this.

**A:** We are referring to the trend (future minus past) which looks like a zonal QSW. We elaborate further in the following paragraph. Changed the word "response" to "trend" in order to avoid confusion

**C**: A quasi-stationary zonal number 5 wave in the northern hemisphere appears to be a prominent feature of the projected circulation trend in high emission scenarios.

R: p.7, line 204 (Fig. 4): (1) It could be helpful for the reader if the authors would include here the underlying past climatology. (2) Does the MMM response really shows a convincing wavenumber 5 signal? There seems to be a low wavenumber signal over eastern asia/western pacific and a higher wavenumber response over North America/Atlantic. A Fourier analysis would probably not show a single peak at wavenumber 5. (3) The authors further say that although this wavenumber 5 response shows up in the MMM (Fig. 4a), the individual models trends represent only low scores if projected onto the EOFs. This seems to suggest that the MMM response has a high score, but is this really the case? Looking at the response wave signal path from North America southward into the Atantic, this does not seem to have a counterpart in the EOFs from Fig. 2. Does the MMM shows a much higher score? (4) Further, I like that the authors included the supplementary Fig. S1 showing all models. But I would have been even more interested in the same kind of Figure, but based on future data as there is a very good agreement between the historical EOFs between models and observations, but there seems to be some stronger discrepancies in the wave response for future projections. So it is not really obvious that the future projections go for a CTP pattern or rather increase in the evolution of separated wave responses as can be seen to some extent already in Fig. S1 for some models in the historical data (EOF2 showing a seperate NA wave pattern?).

A: (1) Added the MMM climatology to Fig. 4

(2) It is true that locally, some parts of the wave are not pure wavenumber 5. But we feel confident to the overall global pattern as a number 5 wave. This is also represented in the Fourier analysis of this exact trend in Simpson et al (2016).

(3) The difference the explained variance between global and regional EOFs is perhaps phrased confusingly. It's not that the trend is similar to the MMM EOFs, but not for individual models. Rather, that using global EOFs is too limiting in terms of capturing the phase of the trend (which is different for different regions). When we split the global EOFs, we get two sets of smaller (but similar looking) EOFs, and the combination of all four allows us to describe the trend. We changed the order of the paragraphs describing this and also added additional explanation (see below).
(4) Added each models EOFs based of RCP8.5 data to supplementary material (Fig. S2)
C: (3) This allows us to use only these regional functions for our analysis (two EOFs for each region). This improvement might seem surprising, considering the spatial similarity between the global and

regional EOFs (see Fig. S2 in the supplementary material). However, this is by virtue of the added degrees of freedom (an index consisting of four regional projections, instead of two global ones).

**R:** p.7, line 207: what exactly is the point that should be tested further? Is it the low projection score?

**A:** This is related to the previous point (3). We want to show that regional EOFs are better for describing the trend.

**C:** In order to confirm that the regional EOFs are more suited for describing the trend, we calculated...

**R:** p.7, lines 213-214: Why does the division into different regions allow to inspect CTPlike responses? Could it not be that doing the calculations for different non-global regions highlights regional wave patterns that are not circumglobal? Fig. 4b shows that the variance of most models cannot be explained captured by the MMM global EOFs, but mainly by EOF 4 (is that correct interpretation?). Wouldn't it be interesting to show this EOF 4? Is it a non global wave pattern of a higher wavenumber? If so, wouldn't that mean that for the future trend the CTP pattern is less relevant and the projections tend to prefer another global wave response? Why isn't that the interpretation of Fig. 4aa and b.

**A:** We hope that this issue is now better explained by our previous changes of pg. 7. We interpret the results in Fig 4 as added degrees of freedom in the phase index. This is strengthened by what we see in daily data – RWPs that correspond to regional EOFs propagate much further (in the AS, nearly circumglobally).

As for the global EOF4, we find it less relevant for 2 reasons: while it helps with visualization, the MMM EOF (constructed as a composite of 36 different EOFs) lacks clear physical meaning. This is because every model has different internal variability. Secondly, the variance of the dataset (lambda) explained by EOF4 is much lower, by design, than that of EOF1&2.

**R:** p.7, line 217 (Fig. S2): What are the values for shading and contours? Shouldn't be the shading in Fig. S2 be identical to the contours in Fig. 3? But blue values increase in Fig. S2 while at the same time red values decrease in spatial extent. Does this mean that the colour contouring is asymmetric between negative nad positive values in Fig. S2?

A: Fixed so that the shading in Fig 3 and S2 is the same.

**R**: p.7, lines 217-219: This is not shown anywhere or is it? The reader hasn't seen the regional EOFs for the obersvational data. So maybe indicating this by sth like "(not shown)".

A: Added "(not shown)" to this sentence.

**R:** - p.7/8, lines 220-224: Any conclusion or interpretation the authors could provide the readers? Comparing the regional to the global EOFs in Fig. S2 I'm surprised thismakes so much of a difference. But the wave pattern for the EOF1 in NA (Fig. S2a) seems to show a wave pattern coming from the south and pointing to the south, suggesting that this wave pattern does not feature a CTP pattern. Could the authors provide some explanation and interpretation to their results. It seems here that the authors now just use the regional EOFs because they better represent the trend of the models. But why is that, is there some possible explanations behind these signals or is this just a useful thing to do to capture the trend, because going to smaller scales usually should allow one at some scale to capture all globaltrends if added together.

**A:** We addressed this point for previous comments. Hopefully our interpretation (more degrees of freedom in an index including four phases) now comes across in the revised version.

**R:** p.8, lines 234-235: (1) I really like the visualization of the data in this phase space diagram of Fig. 5. However, what is the measure of uniformly spread? Starting to look at Fig. 5a this seems a bit strange, as the negative phase of EOF2 does not seem to occur very often for weak EOF1 values, in one of the eight parts there are only 4 blue dots, whereas in others there are more then 20. So it is not really uniformly. Could the authors quantify this to some extent with sth like X % in a 90 degree angle range? That shouldn't be too difficult and this would allow them to make their point more convincingly of how much more the RCP8.5 data is concentrated into a specific region of

the phase space diagram. Probably not necessary, as now I realize this is what Fig. 6 does. (2) Further to this, Fig. 5 does only show one model, right? This should be mentioned by the authors (it is described as all models in the text), so I totally missed this at the beginning.

**A:** (1) We agree that Fig 6 gets across the point of uniformity. (2) We changed the phrasing to emphasize it being single models.

**C:** For all 36 models, historical data is spread approximately uniformly on the phase space (blue dots in Fig. 5, shown for two single model examples)

**R:** - p.8, line 238: Is it obvious that models shows a preference for either NA or AS regions. Is it necessarily "either" for being able to explain why they not project strongly onthe global pattern? Isn't it possible that one model prefer boths, some periods with increased wave patterns over the AS which ressemble the given EOFs there, and some periods with increased wave patterns in the NA region with some arbitrary signal everywhere else. The authors seem to exclude this possibility. What is the reasoning for their conclusion?

**A:** We mention this in the following paragraph: " ... Eight models display a preference in the AS region, with negative (positive) scores for the regional EOF1 (EOF2). Two additional models show these tendencies for both domains". We changed the previous line saying "either" domains to reflect this.

**C**: most models show a preference for certain phases, for either the NA or AS regions (or both, in the case of two specific models).

**R:** p.8, line 244 (Fig. 6): How is the circle calculated? Not clear to me where the assymetry of the circle (right vs left and top vs bottom of the phase space) comes from. This is the standard deviation in physcial space of the individual model monthly data also projected onto the EOFs and then shown in this phase space? If it is the standard deviation of the values in this phase space (as it does sound like in the text), shouldn't it have an equal distant from the origin - or does the circle only seems asymmetrical because of the dashed lines which do not represent an adequate measure of distance? **A:** It is indeed the STD of the values shown in the phase space. The asymmetry is a graphical issue. We fixed this in this version.

C: Fixed the circle shape in the plots and the order of Fig 6 subplots

**R:** p.8, lines 246-247: I'm still confused with everything related to this sigma-measure. Aren't most of the rectangles of RCP8.5 inside the circle? The filled ones represent the same models (the group) in Fig. 6a and 6b, right? But this would mean that most models don't get above 1 sigma for any region or do I misunderstand the plots? Further, isn't the standard deviation calculated from the RCP8.5, so wouldn't we expect most models to be inside of the circle by definition (for simplicity assuming a normal distribution)?

**A:** Your interpretation is correct. Most models do stay inside each circle separately (as can be seen in Fig 6). But when you add up all models from both regions, you get 19 out of 36 models outside (9 in NA, 8 in AS and 2 in both). The wording was a bit misleading, so we changed "most" to "more than half".

**C:** In contrast, more than half of the ensemble's RCP8.5 runs have a mean projection score bigger than 1 sigma for at least one region

**R:** p.8, lines 247-249: Maybe the authors could spend some more time better explain this. I was expecting them to refer to the red rectangles (the mean projection of the RCP8.5 models), but I cannot identify those numbers? First of all, I assume the authors refer to rectangles outside of the circle, but they mention only the sign of the EOF (which would include all of the red rectangles). Further I can identify the eight rectangles outside of the circle in the NA region but not for the AS region (is that because they are on top of each other?).

p.8, line 251: Again, do the authors really mean non-zero or outside of the circle, because their statement is only true for those outside of the circle, isn't it? Following line as well, this is about the rectangles outside of the circle. The 7 cases are then the 6 inside the quadrant and the one with nearly 0 EOF1 very close to it, right? But how can these numbers be associated with the previous

statement of 8 cases with negative (positive) EOF1 (EOF2) for the AS, (representing the second quadrant)?

A: We are indeed referring to the models that are outside of the circle. Some were overlapping so it was hard to tell. We changed the shape to circles with a white outline so that they will be clearer.
C: Interestingly, we can identify a common phasing that is shared by all strongly projecting models in each region (in Fig. 6, these are the filled red rectangles which are outside the circle)

**R:** p.10, line 288: How is it filtered? Difficult to follow in detail, if it is just mentioned that the data is filtered without specifying how.

**A:** Added details about the filtering in the supplementary file.

**C:** ... Two consecutive sequences that are not separated by at least 48 hours are considered one CTP event. Our results were not found to be sensitive to the choice of these parameters. This method of filtering results in some false positive matches. We therefore further filter out all sequences whose composite does not display a wavy signature (alternating mean negative-positive-negative anomalies in the boxes marked in Fig. 8c).

**R:** p.10, lines 290-293: Can the authors say anything about the persistence of the CTP events? p.10, lines 305-309: The authors are measuring also the persistence of the the signal (as this is part of the definition of a CTP event). Would be nice if they could make a statement about the persistence of those events. This should also be sth of interest in terms of climate extreme, because such persistent signal do often lead to extreme events. So knowing about the trend in the persistence could give some hints about the evolution of possible extremes.

A: Added information about the average length of CTP events.

**C:** The average lifetime of the wavepackets does not change between the runs, and was found to be 6 and 5.5 days for IPSL and MIROC, respectively.

**R:** p.10, lines 296-297: Is this really the case? The regions are very different with a shift of partly 60 degrees and huge overlaps. The NA region here captures main parts of both, the Pacific and Atlantic region in Souders et al. (2014). But I assume it depends for what this comparison is used and there is no need for a good agreement between the chosen regions. The authors directly refer to Souder's result of RWP formation, in which case most of the RWP formed in the Pacific region will result in amplified RWPs over the NA region given here, whereas the formation of RWPs in Souder's Atlantic region will contribute to the RWP in the AS region given here - although a direct comparison is not simple because Pacific RWPs could also go all the way towards the AS region whereas RWPs formed in the Atlantic can also decay in the Atlantic.

**A:** That's true. The comparison is not straightforward, so we decided to remove the region comparison and add the comment on one RWP contributing to both groups.

**C**: Note that the comparison to our results is not straightforward. For instance, a RWP formed in the Pacific might be categorized in both of our groups (depending on its path). Additionally, our CTP events likely constitute only a small subset ...

**R**: p.10, lines 297-298: Difficult to compare as Souder's climatology also captures faster propagating RWPs with wavenumbers around 10. So I agree with the authors that Souder's climatology should give a high upper boundary. The waves the authors are interested in (low wavenumbers around 5) should be more comparable with the quasistationary waves investigated in Wolf et al. (2018, Quasistationary waves and their impact on European weather and extreme events). Their Fig. 9 should probably give a lower boundary for the wave occurrences with about 0.6 per season in ERA Interim for very persistent wave signals (minimum 10 day lifetime). So the findings here seem to be well in this range of different climatologies.

**A:** Thank you for bringing this interesting paper to our attention. We now mention this climatology as well in this paragraph.

**C:** Another more specific bound can be found in a work by Wolf et al. (2018). The authors examined the climatology of long-lived quasi-stationary waves using the ERA-Interim dataset, and found an average of ~0.6 instances per winter (DJF) of waves with a lifetime of 10 days or more. This is more

directly related to circumglobal RWPs, which typically require more than a week to complete their path.

R: p.10, line 306: Maybe not using the term "wave activity", which would be strictly speaking a measure of wave strength/amplitude, but here the authors rather mean frequency.A: Changed "wave activity" to "wave frequency"

**R:** p.10, lines 312-313: Why is the RWP at its peak of the life cycle at lag 0? Strictly speaking, one cannot tell this with the applied measure which can only tell its projection peak onto the CTP. But even this probably does not occur at lag 0, or does it? Lag zero is defined as the first of at least 3 consecutive days when the projection value excees the given threshold. But then, would it not be more likely that the peak occurs around day 1 or slightly afterwards?

A: Removed the mention of "peak" and instead, refer to the waves as "somewhat developed".
C: By definition, each such lag 0 day has a high projection score on the preferred phase, meaning that the RWP is already past its initial excitation and is at least somewhat developed.

**R:** p.10, lines 315-316: where does this conclusion come from? What characteristics specifically and how where they affirmed to be realistic? Where does the conlusion for near stationarity comes from? Not clear where the conclusion and description of RWP comes from in this paragraph. Could the author give some more details and explanations for this paragraph?

**A:** These claims are based on the analysis of the composite RWPs. We changed the phrasing to emphasize this and to refer the reader to Figure 10. Also, added an explanation for the stationarity of the waves.

**C**: These composite wave packets (seen in Fig. 10) display realistic characteristics that have been previously affirmed in reanalysis and models ... : Their centers of action remain fairly stationary throughout, meaning that they indeed have near-zero phase speed. This is by construction, as the RWPs were chosen to have persistent strong projections onto the same phase.

**R**: p.11, lines 320-325: How do the author conclude that the wave source is located over Southeast Asia (IPSL case) when there is no statistical significant signal over southeast asia and no clear wave propagation. Further lag 2 and 5 seem to indicate eastward moving phases, contradicting the assumption of a stationary or near stationary signal. This raises again he question if there are not also RWPs with nonzero phase speed. If that would be true, this would mean that individual RWPs could be accounted for more than once (if they include more than one wavelength) which would further be problematic for the identification of the source of this signal or its overall pattern. Further, how many cases are included in this composite? Shouldn't it be something like 1-3 events per season (mentioned last page) for this 93 year period? It is therefore somehow surprising that there is such a strong distortion of the wave signal at positive lags. Do the authors have an idea where this is coming from? Concerning the amplification of the signal, is that not a result of the composite for which the RWPs are forced to have equal phasing around day zero, but no such constraints exists for the days before and afterwards, why one would expect the signal to smooth out and decrease in strength.

**A:** In Figure 10c (IPSL case, -2 lag) there is a statistically significant northerly disturbance over Southeast Asia, which later grows into the RWP itself.

Also, while it is true that the composite phase is not strictly stationary, we make sure to say that the waves have near-zero phase speed. The propagation between figs 10g-10i fits this definition (a few degrees over 3 days).

We agree that the amplification over NA might be an artefact, so we removed this claim from the paragraph. We also now mention the number of RWPs in each composite in the caption for Fig. 10.

R: p.11, lines 340-342: Could the authors explain this more thoroughly how the MM trend (Fig. 4a) can be understood as result of the NA and SA related wave patterns? They add up together to explain the trend given in Fig. 4a? Maybe the authors give some further explanation for the reader.
A: Yes. We now refer the reader to the boxes in Fig. 8 to highlight the specific areas in question.
C: In this light, the spatial structure of the MMM trend can be understood in greater detail. It seems that most anomalous centers can be related to one of the wavetrains that become prevalent in the

projected future in daily data. Specifically, boxes 1-4 (7-9) in fig. 8c correspond to the wave in fig. 10f (10g). Over the Pacific region, it is possible that the North-South dipoles are actually a combination of the two RWP behaviors found in the ensemble (fig. 10g,j).

R: p.11, lines 344-350: What are the implications or the main conclusions here? The main take-away message is that there is a strong jet anomaly upstream of the wave signals? Do the authors have some ideas or hypothesis why the jet should be modified in this way for being able to be associated with a strong wave signal downstream? Nice findings. Do the authors have the information about the connection of their patterns to large scale pattern indices as Fig. 11b looks like a positive NAO (correct?)? Which would be very much in agreement to the findings of Wolf et al (2018, Quasistationary waves and their impact on European weather and extreme events), where they showthat a strong increase in QSW activity along the subtropical jet and the Mediterranean region (EOF2 and 4 in their Fig 11) can be associated with a positive NAO phases. So is the SA group associated with models showing more frequent positive NAO phases? Similar conclusion could be concluded from the PNA, which in its negative phase leads to increased activity of quasi-stationary waves over NA (EOF1 and 3 in Fig 11, wolf et al 2018).

**A:** We've addressed the NAO connection in the major comments section. As for the local jet modification, at this time we still don't have an explanation for this. This will be the focus of our future work in this project.

**R:** p.11/12, lines 351-355: This is rather a question, referring again to a point mentioned earlier: are the authors sure about the compositing of RWP events (is there no double counting for lagged RWPs)? Because those double counting with time lags could obscure the temporal relation between tropical forcing and the associated RWP.

**A:** It is true that the connection to tropical forcing might be obscured (we address this in the Discussion chapter). However, we don't think that double counting is the reason for it. Based on how we define the CTP events, we are confident that phase-propagation in the composite in negligible.

**R:** p.12, lines 369-370: But this was not working for the projection onto the global CTP, was it? This result was associated with the separation of the signal into different regions. So why can the results be associated with the evolution of the overall CTP? It is not obvious why the results for the analysis, restricted to specific regions, should explain the behaviour of a circumglobal teleconnection pattern. Could the authors maybe make this clearer?

A: As we tried to clarify in the open discussion, we use the term CTP in a broad sense – a class of synoptic, zonal number 5 patterns which also have a clear signature in larger scales (subseasonal, as well as hemispherical). The CTP is fundamentally tied to local scales, as there is no instance of a CTP wave instantaneously encompassing the entire globe. We've tried to stress this point more in the revised version, as seen below:

**C:**

Line 101 - Despite the "Circumglobal" part of its name, throughout this work we use the term CTP in its most general sense - a class of regional wavenumber-5 patterns, rooted in the synoptic scale while showing a clear signature in the subseasonal range and over the entire hemisphere. Line 67 - This makes the CTP a unique bridge between timescales. The prevalence and quasi-stationary nature of these synoptic RWPs allow their influence to manifest on a subseasonal scale as well. Line 412 - The primary finding of this work is that the majority of GCMs project the CTP to develop a preferred longitudinal phasing over time. While this change is local in nature, its effect is seen on larger scales (both spatially and temporally).

R: p.13, lines 418-420: This link between mean flow (waveguide) and the resulting wave pattern seems indeed crucial. The authors show the spatial relationship which seems to show increased/shifted jet stength upstream of the onsetting wave pattern. Do the authors have some interpretation or insight into the dynamical link for this connection, or is this just a result of increased jet strength (shifted away from) in the locations were are climatologically seen less activity of occurring wave patterns (for those regional wave groups)? A: See above.

**Referee #2:**

**Major Comments:**

**R**: Are the EOFs for DJF the same as for each month separately? Is there one Month that dominates the seasonal signature? For reference, Ding and Wang 2005 showed that the CTP had different signatures throughout JJA.

**A:** Monthly EOFs are very similar, for both reanalysis and the MMM. The December EOFs are slightly shifted equatorward, but we do not expect this to impact our results.

**R:** Could the authors quantify which models have been most accurate in the representation of the CTP compared to reanalysis? In the light of the 'strong disagreements'

**A:** Added a supplementary plot showing the correlation scores between each model's EOFs and the reanalysis (Fig. S4) between models, this might allow some careful statements which model is more reliable in terms of future projections

**R:** Could the authors expand on what intrinsic mid-latitudinal mechanisms might trigger and maintain the CTP and its preferred phase?

**A:** We mention several mechanisms for CTP excitation in the Introduction section. In the revised version we now delve deeper into the connection between the CTP and the NAO specifically (see subsection 3.4, the Discussion section and Fig. 12). Other possible triggers (like recurving cyclones or stratospheric disturbances) are beyond the scope of our project, and so we focus on NAO and tropical convection.

As for the maintenance of preferred phases, we still don't have an answer unfortunately. As we state in the conclusion of the paper, we will have to explore this question using idealized modelling in the future.

**Minor Comments:**

**R:** -Include 'wintertime' in title. Further 'teleconnections' refer to patterns such as 'ENSO' or 'MJO' could this be further specified in the title to avoid confusion? -(p.1, l.2)

**A:** It is specified that these are subseasonal teleconnections, so there will be no confusion with ENSO or MJO. We added 'wintertime' to the title.

R: '..variability ARE upper tropospheric..' -(p.1,l.4)

A: "is" is referring to "a common feature of...", so this is not a mistake.

**R:** Others have used the abbreviation CGT (see e.g. Ding &Wang 2005), consider changing CTP to CGT to stay consistent with the terminology used in the literature.

**A:** This teleconnection has several widely used names, unfortunately. Some use CGT, some use CWP (Haarsma & Selten, 2012; Risbey et al, 2015) and some CTP (Yuan et al 2011, Dai et al 2017). We prefer CTP in order to be consistent with a previous paper (Harnik et al 2016)

**R:** -Try to avoid effusive / inessential expressions such as 'dramatically' (p.1. l.1), 'surprisingly' (p.1 l.4), (p.3. l.78) and (p.12 l.381), 'first described two decades ago' (p.2,l.25) 'most definitely' (p.14 l.437), 'unsurprisingly' (p.9 l270), 'most definitely' (p.14, l.437)

**A:** We toned down the language for the given examples: 'dramatically' to 'considerably'; 'surprisingly' to 'seemingly'/'actually'/'unexpected'; 'unsurprisingly' to 'as expected'; 'most definitely' to 'indeed'

**R:** -Could the authors add a few sentences on differences to summer Circumglobal / stationary waves to the introduction?

**C**: In this work we focus on the wintertime CTP, but a summertime variant exists as well. In the boreal summer, the NH jet stream shifts poleward and is typically weaker. Therefore, the stationary waves associated with the CTP are shorter in scale (mostly k=6) and lifetime (Teng & Branstator, 2017). Nevertheless, the summertime CTP was also found to be related to extreme weather, such as

heat waves in Southeast China (Wang et al, 2013) or extreme precipitation over western Europe (Saeed et al, 2014).

**R:** -(p.1, l.4) Maybe change 'likeliness .. emerges' their 'frequency increases' or similar.

**A:** We mention later in the abstract (where we discuss our results) that RWP frequency increases. However, in line 4 we are posing the question that our work tries to answer – why is it that the trend looks like the CTP.

R: (p.1, l.6) Name the timescales (Monthly and 3-day mean right?)

C: We attempt to elucidate this link across timescales (daily, monthly and climatological), ...

**R:** -(p.1,l.11) 'This categorization strongly corresponds to the ensemble spread in local trend magnitude.' It is not clear to me what this means in this context.

A: Changed the wording to clarify

**C**: The ensemble is thus divided into subgroups based on region of increased wave activity. For each model, this region corresponds to a more pronounced local trend, which helps explain the ensemble projection spread.

**R:** -(p.1 l.15) 'Thus, we conclude that this hemisphere-wide climate change signature is actually comprised of several regional effects'. –What hemisphere wide climate change signature? Better use 'response'. Also the authors highlight in the paragraph before, that changes are found visible on a more regional level, how is it a hemisphere-wide signal? Please be a bit more concise

**A:** The climate response is hemispherical - a single zonal number-5 wave trend. This wave however is a combination of several regional signals. We now reflect this point in the revised abstract.

**C:** Their likeness seemingly emerges as a robust signal in future meridional wind trend projections in the Northern Hemisphere, which take the form of a zonal wave encompassing the midlatitudes. ... Thus, we conclude that this climate change response, seemingly a single large-scale wave, is actually comprised of several regional effects which are related to shifts in CTP phase distributions.

**R:** (p.1, l.20) 'Projections of future circulation trends, driven by anthropogenic climate change, commonly display large scale patterns.' It feels like this statement requires a reference. **A:** Added a reference to the IPCC aAR5

**R:** (p.1 l.21) '..in order to provide dynamical reasoning and theory.' In order to test hypotheses and theories?

**C**: Studying these structures in the context of changes in the climatological mean flow is essential in order to understand the underlying basic dynamical mechanisms.

**R:** p.1 l22.) '..development on finer scales?' What scales, higher temporal resolution? **A:** Changed to "a higher temporal resolution"

**R:** (p.1 l22.) 'changes in subseasonal to seasonal fluctuations' – changes in variability or changes in subseasonal circulation patterns?

**A:** Both are applicable. We believe this can be stated generally as this is just an introductory phrase.

R: (p.2 l.26.) 'term' change to 'pattern'

**A:** We are referring to the term CTP as describing waves, so "pattern" doesn't work in this context in this sentence.

**R**: (p.2 and later) The 'CTP' is described as 'a pattern' and then as 'the wave' or and then as 'waves'. Later it is described as 'a class of related patterns', all of them 'waves'. It would be helpful if the authors could rewrite that part while being more precise in terminology. The sentence in I. 37 should come a bit earlier to clarify the hierarchy among the terms, which are seemingly used synonymous earlier in the paragraph.

A: We changed the phrasing to better indicate that we are talking about several possible waves ('the wave circumscribes' was changed to 'each wave circumscribes'), and also moved up the sentence regarding the CTP being a class of patterns.

**C:** The term describes quasi-stationary Rossby waves in the upper-troposphere, which are zonally oriented and span the Northern and Southern Hemispheres (NH and SH respectively). On a

subseasonal scale, one can picture the CTP as a "family" of related patterns, all of them waves with an arbitrary longitudinal phase. As they are quasi-stationary ...

R: (p.3. 170) is it an acceleration or a poleward shift (or both)?

A: Simpson et al are specifically referring to the acceleration of the jet (larger mean U).

**R:** (p.23, l.74) works -> studies

A: Changed 'works' to 'studies'

**R:** (p.3 I.79) please further specify what the conceptual gap is.

**C:** There is a conceptual gap that complicates the establishment a direct causal link between the CTP and the wavy number-5 trend found in climate change projections. Namely, how do changes in jet driven subseasonal variability translate to long-term climatological shifts?

**R:** (p.4 l.98) do the patterns depend on the chosen mid-lat range?

A: No. You can choose a narrower range and you will get the same pattern.

R: (p.4 l.105) this sentence seems grammatically wrong?

A: We find no grammatical mistakes in this sentence.

R: (p.5 l. 143) over which years is the climatology defined for reanalysis datasets?

**A:** We don't present monthly deviation projections for reanalysis, so we don't define a climatology for it

**R:** p.5 l. 148 ff)Wouldn't a negative projection score mean a preferred phase opposite to the one in question while a score of zero would refer to an arbitrary phase?

**A**: A negative projection score is indeed the antiphase, but a zero score doesn't tell us anything about the phase since the data might not even look like a wave. We explain this point in the revised version

**C:** Zonal number 5 patterns will have a large absolute score (1 for a perfect copy, -1 for the same wave in antiphase), and a zero score represents some flow unrelated to the CTP.

R: (p.6 l.159) why is a running mean of three days chosen?

**A:** We wanted to smooth out some noise out of the daily data and chose a window that is shorter than the average CTP lifespan. Our results are not sensitive to this choice.

R: (p.6 l.163) 'future' (p.6 167) ; (p.9 l.264) future , past

A: These are referring to periods we named 'Future' and 'Past', so we capitalize it throughout

**R:** (p.6 167) OLR – provide full expression before using an Acronym, here: outgoing longwave radiation?

A: fixed

**R:** p.6 l171) What are the signatures of the other EOFs? Are they more local and excluded from the analysis for that reason?

**A:** Some of the EOFs are some phase shifted version of the circumglobal wave and others are a combination of smaller-scale waves. They are excluded from the analysis since they explain very little of the variability (single digit lambdas).

**R:** (p.7. l. 220) Please be more specific, I don't understand this sentence.

**C:** In order to confirm that the regional EOFs are more suited for describing the trend, we calculated how much of its spatial variance was explained by the EOFs (Fig. 4). This was done individually for every model, as well as for the multi-model mean (using composite EOFs and the MMM trend). For most models used, the variance is spread quite evenly across the first five EOFs ...

**R:** (p.9 l.275) what is meant by temporal frequency here, their occurrence on subseasonal timescales?

**A:** Yes. The stationary signature in monthly data is the average of multiple RWPs, so we don't know when they occur.

R: p.9 l.281) can this statement be quantified?

A: Added the daily-monthly preferred phase difference

**C**: The daily phase preferences are close to the monthly ones in terms of angle (|phiday-phimon|<pi/4)

**R:** (p.9 l.286) as a three-day running mean was applied it is incorrect to speak of days in this context. Better use 'timestep' or similar.

A: Changed 'days' to 'daily timesteps'

R: (p.10. l.288) How are events filtered?

A: Added an explanation in the supplementary materials

**C:** We define "CTP events" in daily mean data. ... Two consecutive sequences that are not separated by at least 48 hours are considered one CTP event. Our results were not found to be sensitive to the choice of these parameters. This method of filtering results in some false positive matches. We therefore further filter out all sequences whose composite does not display a wavy signature (alternating mean negative-positive-negative anomalies in the boxes marked in Fig. 8c).

R: (p. 10 l. 290) change 'observational' to 'reanalysis' (here and everywhere else)

A: Changed to 'reanalysis' where relevant

R: (p.10 l. 299) 'much along the lines' -> similar to

A: This is a stylistic choice. We believe the intention is clear in this sentence

**R:** (p.11 I.340) The conclusion is hard to understand, could this be re-formulated?

A: Rephrased the conclusion and added a reference to Fig 8 to highlight specific regions

**C:** In this light, the spatial structure of the MMM trend can be understood in greater detail. It seems that most anomalous centers can be related to one of the wavetrains that become prevalent in the projected future in daily data. Specifically, boxes 1-4 (7-9) in fig. 8c correspond to the wave in fig. 10f (10g). Over the Pacific region, it is possible that the North-South dipoles are actually a combination of the two RWP behaviors found in the ensemble (fig. 10g,j).

**R:** In the Discussion / Conclusion section: Could the authors provide the Figures in which each of the discussed findings is shown?

A: Added figure references in this chapter

R: (p.12 l.363) change 'business as usual' to 'high emission'.

**C:** the latter represented by the high emission RCP8.5 scenario

R: (p.12 I.364) what does 'decent skill' mean in this context?

**A:** This means that most models capture the centers of action of the wave and the general phase of the leading EOFs but the patterns are not identical to reanalysis. Therefore, we can at least talk about a CTP phenomenon in each model

**R:** (p.12 I.369) add a short statement on consquences for predictability / future surface weather. **A:** We briefly mention the possible effects on regional precipitation in lines 430-435. We now added some more detail to highlight this issue. On predictability we don't have a statement to add to the conclusions.

**C:** It is left for future works to examine the possible relations between the CTP and these regional precipitation trends in CMIP models. However, we hypothesize that these results would have implications on surface weather, due to the presence of these recurring, persistent types of flow.

R: (p.12 I.379). Where are those regions?

A: Added the increased initiation areas.

**C:** ... such as areas of prominent RWP initiation (North Atlantic, and the Western and Central Pacific)

R: (p.12 I.383) Reference?

**A:** Which sentence should be referenced? The statement 'Both are associated with regional precipitation anomalies over their respective domains' is based on the two citations in the previous sentence.

R: (p.12 I.385) 'Seemingly'? Does it or doesn't it?

**A:** We can't positively say that there is or isn't a contradiction, but we are raising this as a hypothetical issue and immediately after say that future work will be done on the subject.

R: (p.13 l.397) 'However: ::' I don't understand this sentence.

A: We moved this sentence to the next paragraph and rephrased it to be clearer

**C:** However, the relationship between seasonal, subseasonal and synoptic timescales is not straightforward.

**R:** (p.13, l.406) Another important scale is the spatial one'. Consider removing this sentence. **A:** Removed the sentence and joined the two paragraphs together.

R: (p.14 I.435) 'There is difficultie in singling out..' Could the authors be more specific?

A: Rephrased to make It clearer

**C**: The CTP is only one mode of many which influence wintertime variability in the midlatitudes, and it's difficult to isolate its underlying mechanisms in the context of a fully coupled GCM.

R: (p.14, l.437) provide reference

A: Added a citation of Teng & Branstator 2017.

**R:** What else could provide relevant forcing? Consider citing Garfinkel et al. 2020 (https://journals.ametsoc.org/doi/10.1175/JCLI-D-19-0181.1?mobileUi=0)

**A:** This article is very interesting, but it's difficult to adapt their conclusions (that deal with planetary scale stationary waves) to the subseasonal-to-synoptic waves of the CTP. We expanded our discussion of forcing by including the link between NAO phases and CTP-related RWPs (see Section 3.4 and the Discussion chapter for more detail).

[revised manuscript text omitted]

- 180 phase index doesn't have physical meaning. Our results are not sensitive to the choice of percentile thresholdwe average all remaining monthly values to obtain the mean projection score of every experimental run. Patterns with an overall arbitrary phasing will have oppositely signed monthly scores cancelling each other, and therefore a mean projection close to the origin. When a model's mean score exceeds the RCP8.5 Multi-Model Mean (MMM) projection score by one standard deviation, it is considered to have a "preferred" phase. A model subgroup, used for trend and forcing comparisons, is defined by all models
- 185 whose preferred phase over a geographical domain (NA or AS) occupy the same half-plane on the phase space.

Statistical significance of group composites is determined by a sign test, which indicates where a certain percentage of group members have the same sign as the composite mean. The chances for a given percentage of events to have the same sign as the composite mean are determined using a binomial formula, assuming equal chances for positive and negative anomalies.

**2.4 CTP events**

We define "CTP events" in daily mean data, which are essentially a synoptic manifestation of the wave patterns that are found in monthly data. We capture quasi-stationary Rossby wave packets that are in the same domain and. These are essentially Rossby Wave Packets that are nearly in phase with the preferably phased monthly patterns (only on the 500 hPa level, as explained previously). To do so, we first 
[revised manuscript text omitted]

---

## Author Response (AR2)

**Point by Point Response & Changes to Manuscript**

**E** – Referee comment ; **A** – Author response ; **C** – Manuscript change

**Minor Comments:**

**E:** I think it is satisfactory to use v winds rather than the wind perpendicular to the waveguide in this case, but perhaps you could mention the approach of Wolf et al as an alternative.

**C:** This approach is focused on the phase of the RWP, but the use of EOFs also captures the meandering character of their path. Alternatively, one can possibly further filter the wind field and use only the component perpendicular to the background flow (Wolf & Wirth, 2017).

**E:** It would be good to clarify in the main text in which cases EOFs 1 and 2 are / are not clearly separated from EOF 3, as this seems quite useful (just in the NCEP 1 data?).

**A:** We now explicitly state which datasets have separated EOFs. This is also mentioned for individual models in Table 1.

**C:** For ERA-Interim, the two EOFs are not well-separated from one another (as well as from the third EOF) according to the definition set by North et al (1982). For NCEP-I, a longer dataset, the three leading patterns are well-separated from each other. …

Of the 36 ensemble members, 28 have their two leading EOFs well-separated from the third function (see Table 1).

**E:** I didn't find figure S4 in the supplementary material. Please check this and link to it from the main text.

**A:** Added the figure

**E:** I would reconsider the use of the word 'bias' throughout the paper, as it seems to be used more generally to refer to a difference between models than is common in the literature. I would restrict bias to the mean discrepancy between model and observations over a defined period.

**A:** Rephrased all mentions of 'bias' to 'difference from the MMM'

**E:** typos - Line 104: brought about by changes; Line 285: non-zero; Line 329: missing date in reference

**A:** fixed

[revised manuscript text omitted]